# Guideline-Informed MLLM Reasoning for Pathology-Aware Postoperative Prostate CTV Segmentation

Yinhao Wu [1]  ⓘD                YXW2120@MAVS.UTA.EDU
Hengrui Zhao[2]  ⓘD            HENGRUI.ZHAO@UTSOUTHWESTERN.EDU
Haiqing Li[1]  ⓘD                  HXL9110@MAVS.UTA.EDU
Wenliang Zhong[1]                WXZ9204@MAVS.UTA.EDU
Hehuan Ma[1]                 HEHUAN.MA@MAVS.UTA.EDU
Yuzhi Guo[1]                  YUZHI.GUO@MAVS.UTA.EDU
Dan Nguyen[2]          DAN.NGUYEN@UTSOUTHWESTERN.EDU
Daniel Yang[2]           DANIEL.YANG@UTSOUTHWESTERN.EDU
Steve Jiang[2]  ⓘD         STEVE.JIANG@UTSOUTHWESTERN.EDU
Junzhou Huang[1]  ⓘD          JZHUANG@EXCHANGE.UTA.EDU

[1] *The University of Texas at Arlington, Arlington, TX, USA.*

[2] *UT Southwestern Medical Center, Dallas, TX, USA.*

**Editors:** Accepted for publication at MIDL 2026

## Abstract

Accurate segmentation of the Clinical Target Volume (CTV) is a critical prerequisite for precise radiotherapy planning, pursuing complete irradiation of microscopic disease while minimizing toxicity to surrounding healthy organs. However, achieving automated CTV segmentation remains highly challenging due to the invisible microscopic disease on planning CT and the necessity of incorporating clinical context into delineation decisions. Unlike previous methods that rely solely on visual features or coarse global text reasoning, we propose **ReaCT**, a unified framework that reformulates CTV segmentation as a multimodal reasoning task by explicitly integrating pathological information with visual context. Specifically, we introduce a Guideline-Informed Attribute Extractor that follows the information-retrieval workflow of radiation oncologists. By distilling knowledge from clinical guidelines, this module filters and structures lengthy pathology reports into a concise set of clinically determinative pathological attributes, effectively bridging the semantic gap between unstructured clinical records and segmentation networks. Furthermore, we develop an Attribute-Specific MLLM Reasoner built upon a 3D residual U-Net that performs fine-grained spatial reasoning. By leveraging a sequence of attribute-specific query tokens, the model disentangles the distinct target implications of individual pathological attributes, enabling fine-grained anatomical alignment via multi-scale fusion using Two-Way Transformers. Experiments on a postoperative prostate cancer dataset demonstrate that ReaCT achieves state-of-the-art segmentation performance and exhibits strong robustness, with pronounced improvements under limited-annotation settings.

**Keywords:** Clinical Target Volume, Radiotherapy Planning, 3D Image Segmentation.

## 1. Introduction

Radiotherapy is one of the most common treatments for cancer, delivering radiation doses to the target volume while sparing surrounding healthy tissues (Bi et al., 2019; Liu et al., 2021). Achieving the optimal therapeutic effect relies on precise treatment planning, including the

delineation of the Clinical Target Volume (CTV) for microscopic tumor extensions (Lee et al., 2018; Balagopal et al., 2021). Different from Gross Tumor Volume (GTV) which usually have a distinct contrast on CT images, CTV can be invisible on planning CT images and its segmentation presents a formidable challenge. This difficulty is exacerbated in postoperative radical prostatectomy, where the surgical removal of the prostate and nearby tissues leaves a void in the target area, and the CTV boundaries are usually invisible. This necessitates complex reasoning from clinical context in addition to visual perception.

To facilitate consistent and reliable delineation, radiation oncologists typically need to integrate patient-specific pathological attributes derived from pathology reports alongside planning CT images in their decision-making process. These attributes are essential because they must be interpreted together with consensus clinical guidelines to determine the appropriate boundaries of the CTV (Jansen et al., 2000; Chang et al., 2007). For instance, different pathological stages determine whether only the proximal base of the seminal vesicles should be included, whereas confirmed seminal vesicle invasion mandates the inclusion of the entire seminal vesicle bed (Dal Pra et al., 2023). Consequently, distinct from conventional segmentation tasks, CTV segmentation inherently requires the incorporation of multimodal knowledge to reason about regions susceptible to microscopic metastases that are indistinguishable based on visual features alone.

Recently, Multimodal Large Language Models (MLLMs) have demonstrated exceptional reasoning capabilities when handling implicit or abstract text instructions (Lai et al., 2024; Zou et al., 2025). Drawing inspiration from these advancements and considering the intrinsic reasoning demands of CTV segmentation, we posit that empowering MLLMs to perform multimodal reasoning is essential for resolving the ambiguity of invisible target boundaries. However, seamlessly integrating such specialized knowledge into MLLM architectures presents significant challenges. First, patient records such as pathology reports and clinical notes are typically lengthy, unstructured, and rich in domain-specific terminology, making it difficult to effectively encode and align them with segmentation networks. Second, existing MLLM-based approaches typically rely on a single, coarse reasoning token, which limits their ability to capture the fine-grained correspondence between individual pathological attributes and their distinct spatial implications for CTV segmentation. A more comprehensive review of related work is provided in Appendix A.

To address the above limitations, we propose **ReaCT**, a unified multimodal framework that reformulates CTV segmentation as a reasoning-driven task by integrating patient-specific pathological attributes with visual context. As illustrated in Figure 1, ReaCT incorporates a Guideline-Informed Attribute Extractor to follow the information-retrieval workflow commonly used by radiation oncologists. This module first distills relevant radiotherapy consensus guidelines to derive a principled set of pathological attributes that govern the spatial extent of the CTV. Leveraging this guideline-grounded schema, it then processes raw pathology reports through a multi-stage pipeline involving keyword-based context retrieval, semantic verification, and value standardization, transforming unstructured clinical documentation into a concise and structured attribute set for downstream multimodal reasoning. Subsequently, ReaCT introduces a multimodal CTV segmentation network built upon a 3D residual U-Net backbone and a custom MLLM reasoner. The MLLM Reasoner jointly processes visual and textual tokens together with a sequence of attribute-specific <SEG> query tokens. This granular design enables fine-grained reasoning,

where each query token independently encapsulates the distinct target implications of a specific pathological attribute by aggregating relevant multimodal context. The hidden embeddings from the last layer corresponding to these `<SEG>` tokens are aggregated and then fused with multi-scale visual features through bi-directional transformer modules at each decoder stage to ensure precise anatomical alignment. Benefiting from this fine-grained multimodal fusion, ReaCT achieves state-of-the-art performance on postoperative prostate cancer datasets and exhibits pronounced robustness even in limited-annotation regimes. We highlight the following contributions:

- We propose **ReaCT**, a unified framework that reformulates CTV segmentation as a multimodal reasoning task by explicitly integrating patient-specific pathological attributes with visual context. This formulation addresses the inherent clinical need for multimodal integration and provides a principled mechanism toward anatomically and clinically coherent CTV segmentation.

- We design a Guideline-Informed Attribute Extractor that follows the information-retrieval workflow used by radiation oncologists. By distilling knowledge from consensus guidelines, the extractor transforms lengthy pathology reports into a concise set of clinically determinative attributes, bridging the semantic gap between unstructured clinical records and downstream segmentation networks.

- We develop an Attribute-Specific MLLM Reasoner that performs fine-grained spatial reasoning through a sequence of query tokens, enabling the model to disentangle the distinct target implications of individual pathological attributes and ensuring precise anatomical alignment even in limited-annotation regimes.

## 2. Methodology

As illustrated in Figure 1, ReaCT comprises two parts. First, a Guideline-Informed Attribute Extractor emulates the expert information-retrieval workflow, distilling consensus guidelines to transform unstructured pathology reports into a concise set of determinative attributes. Second, a multimodal CTV segmentation network predicts the CTV mask by integrating the 3D CT volume with patient-specific attributes. Built upon a 3D U-Net and an Attribute-Specific MLLM Reasoner, ReaCT utilizes a sequence of query tokens to generate fine-grained spatial reasoning embeddings, which are fused into the decoder via multi-scale Two-Way transformers to ensure precise anatomical alignment.

### 2.1. Guideline-Informed Attribute Extractor

To bridge the semantic gap between unstructured clinical records and the downstream segmentation network, we construct a Guideline-Informed Attribute Extractor based on GPT-4o (Hurst et al., 2024). This module is designed to reflect the information-retrieval process used by radiation oncologists through a two-step procedure: (1) distilling relevant knowledge from consensus radiotherapy guidelines into a compact attribute schema, and (2) extracting patient-specific attribute values from raw pathology reports.

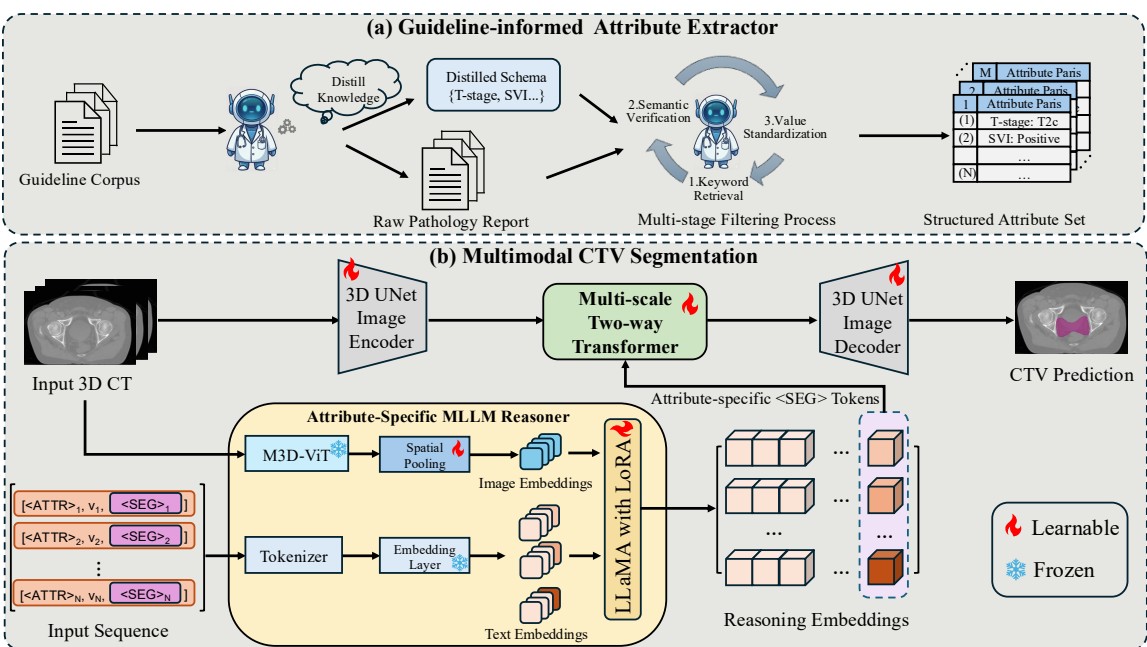

Figure 1: The overall framework of ReaCT. (a) A Guideline-Informed Attribute Extractor emulates clinical information-retrieval workflow by distilling knowledge from consensus guidelines to obtain structured determinative attributes. (b) A Multimodal CTV Segmentation Network integrates a 3D U-Net with an Attribute-Specific MLLM Reasoner, fusing fine-grained reasoning embeddings derived from distinct query tokens into the decoder via multi-scale Two-Way Transformers to ensure precise anatomical alignment.

### 2.1.1. GUIDELINE-BASED SCHEMA CONSTRUCTION

To establish a guideline-grounded reasoning framework for attribute extraction, the proposed extractor first constructs a comprehensive guideline corpus $\mathcal{G} = \{g_m\}_{m=1}^{M}$ by retrieving and aggregating consensus radiotherapy guidelines, such as ESTRO (Niyazi et al., 2016), NCCN (Carroll et al., 2016), and RTOG (Kruser et al., 2019). Subsequently, under a fixed prompting policy $\mathcal{P}_{\text{schema}}(\cdot)$ with deterministic decoding, the extractor employs the LLM $\Psi_{\text{LLM}}$ to analyze the retrieved documents and distill pathological factors that explicitly influence the CTV definition. Formally, this distillation process yields a principled attribute schema

$$\mathcal{S} = \Psi_{\text{LLM}}\big(\mathcal{P}_{\text{schema}}(\mathcal{G})\big) = \{k_i\}_{i=1}^{N},$$

where each $k_i$ corresponds to a clinically determinative pathological attribute.

### 2.1.2. MULTI-STAGE ATTRIBUTE EXTRACTION.

Guided by the schema $\mathcal{S}$, the extractor processes the raw patient-specific pathology report $\mathbf{D}_{\text{raw}}$ to derive a structured set of attribute–value pairs. To mitigate the noise and redun-

dancy inherent in unstructured medical text, a coarse-to-fine filtering pipeline is applied. First, a keyword matching operator $\mathcal{F}_{\text{key}}(\cdot)$ is used to retrieve a subset of relevant text segments $\mathbf{T}_{\text{rel}} \subset \mathbf{D}_{\text{raw}}$ that potentially contain information linked to $\mathcal{S}$, thereby efficiently narrowing the search space. Subsequently, the extractor functions as a semantic verifier $\Phi_{\text{verify}}(\cdot)$ on $\mathbf{T}_{\text{rel}}$, filtering out irrelevant narratives (e.g., unrelated medical history) and eliminating redundant statements, while retaining only textual evidence that directly informs the attributes in $\mathcal{S}$. Finally, the verified context is aggregated to assign a standardized value $v_k$ for each attribute $k$. This process produces a structured clinical attribute set

$$\mathcal{A} = \{(k, v_k) \mid k \in \mathcal{S}, v_k \neq \varnothing\}.$$

The curated attribute set $\mathcal{A}$ serves as textual input for the downstream multimodal reasoning module, enabling more accurate and context-aware CTV segmentation. Detailed implementation is provided in Appendix B.

## 2.2. Multimodal CTV Segmentation

The segmentation network consists of a 3D U-Net-based segmentation path that encodes and decodes multi-scale spatial features, and an MLLM reasoner that generates fine-grained reasoning embeddings from the clinical attributes and corresponding image context. In this work, we use the term *reasoning* to denote guideline-driven conditional inference over spatial decisions, rather than static multimodal conditioning via feature fusion. Unlike conventional multimodal approaches (Liu et al., 2023; Zhao et al., 2025a,b) that model correlations of the form $P(\text{Mask} \mid \text{Image}, \text{Attributes})$ through concatenation or cross-attention, ReaCT performs sequential, attribute-conditioned inference within a unified semantic space. Specifically, ReaCT interleaves visual tokens and pathological attribute tokens into a single autoregressive multimodal sequence $X = \{x_t\}_{t=1}^T$, where each token corresponds to either a visual embedding, an attribute embedding, or an attribute-specific reasoning query. Through deep self-attention across transformer layers, textual tokens representing clinical rules repeatedly interact with visual tokens, enabling the model to learn conditional dependencies of the form $P(x_t \mid x_{<t})$, where each token attends jointly to prior visual context and attribute-specific cues that guide spatial delineation decisions.

### 2.2.1. Segmentation Path

We adopt a 3D residual U-Net (Çiçek et al., 2016) as the visual backbone to facilitate the hierarchical interaction between anatomical features and clinical reasoning. The encoder $\mathcal{F}_\phi$ first processes the input CT volume $\mathbf{x}_{\text{img}} \in \mathbb{R}^{B \times 1 \times H \times W \times D}$ to extract a pyramid of multi-scale spatial features $\{\mathbf{f}_l\}_{l=1}^L$, where $\mathbf{f}_l \in \mathbb{R}^{B \times C_l \times H_l \times W_l \times D_l}$ denotes the feature map at the $l$-th resolution scale. Meanwhile, the proposed MLLM Reasoner jointly encodes the visual tokens and the extracted pathological attributes to generate a set of fine-grained reasoning embeddings $\mathbf{H}_{\text{reason}} = \{\mathbf{h}_k\}_{k=1}^N \in \mathbb{R}^{B \times N \times d'}$, where each $\mathbf{h}_k$ captures the inferred spatial implication of the $k$-th pathological attribute. To integrate the fine-grained reasoning embeddings $\mathbf{H}_{\text{reason}}$ generated by the MLLM into the decoding path, we employ the Two-Way Transformer fusion mechanism from the Segment Anything Model (SAM) (Kirillov et al., 2023). This module facilitates bidirectional interaction between the spatial features $\mathbf{f}_l$ and

attribute queries $\mathbf{H}_{\text{reason}}$ at each upsampling stage, yielding a clinically modulated representation $\tilde{\mathbf{f}}_l = \text{TwoWayBlock}(\mathbf{f}_l, \mathbf{H}_{\text{reason}})$. Finally, the decoder $\mathcal{D}_\phi$ progressively upsamples and aggregates these fused features $\{\tilde{\mathbf{f}}_l\}_{l=1}^L$ to reconstruct the segmentation mask, outputting the final CTV probability map $\hat{\mathbf{y}} \in [0,1]^{B \times 1 \times H \times W \times D}$.

### 2.2.2. Attribute-Specific MLLM Reasoner

We design the MLLM reasoner to jointly encode 3D image features and the extracted pathological attributes, enabling reasoning-guided CTV segmentation. Specifically, we build upon M3D-LaMed (Bai et al., 2024), a specialized multimodal large language model for 3D medical imaging, which consists of a 3D ViT image encoder (M3D-ViT), a spatial pooling projector, and a LLaMA-2-7B (Touvron et al., 2023) language backbone.

**Multimodal LLM for Fine-Grained Anatomical Reasoning.** Given the 3D CT volume $\mathbf{x}_{\text{img}}$, the pretrained M3D-ViT encoder $\Phi_{\text{ViT}}$ first extracts patch-level embeddings $\mathbf{Z}_{\text{img}} = \Phi_{\text{ViT}}(\mathbf{x}_{\text{img}}) \in \mathbb{R}^{M_0 \times d_v}$, where $M_0$ denotes the number of 3D patches and $d_v$ represents the vision hidden dimension. To align with the LLM latent space, these embeddings are processed by a spatial pooling projector $\mathcal{P}_\psi(\cdot)$, which applies 3D average pooling followed by a series of Multi-Layer Perceptrons (MLPs). This projection yields the compressed visual embedding $\mathbf{F}_{\text{img}} = \mathcal{P}_\psi(\mathbf{Z}_{\text{img}}) \in \mathbb{R}^{M \times d'}$, where $M = 128$ is the reduced token length and $d'$ aligns with the hidden dimension of the LLM.

To achieve fine-grained anatomical reasoning, we augment the tokenizer's vocabulary with a set of learnable attribute-specific tokens $\{\texttt{<SEG>}_i\}_{i=1}^N$. Each token $\texttt{<SEG>}_i$ is designed to act as a dedicated reasoning query for the $i$-th attribute in $\mathcal{A}$. Formally, let $\mathbf{A}_i$ denote the tokenized embedding of the attribute pair $(k_i, v_i)$ (e.g., [SVI, Positive]). We construct the joint multimodal input sequence $\mathbf{X}$ by concatenating the visual context with a series of attribute-reasoning blocks. Formally, for the $i$-th attribute, the input sequence $\mathbf{X}_i$ is:

$$\mathbf{X}_i = \big[\mathbf{F}_{\text{img}}; \mathbf{A}_i; \texttt{<SEG>}_i\big]. \tag{1}$$

Each input sequence functions as an independent reasoning unit, prompting the model to synthesize the shared visual context $\mathbf{F}_{\text{img}}$ with the specific pathological attribute $\mathbf{A}_i$ to encode the spatial intent into the corresponding $\texttt{<SEG>}_i$ token.

Subsequently, the input sequence $\mathbf{X}$ is processed through the LLM backbone $\mathcal{M}_\theta$ with $L$ transformer layers to model the conditional dependency $p(x_t|x_{<t})$, where $t$ denotes the token index. The hidden states are recursively transformed as:

$$\mathcal{H}^{(\ell)} = \mathcal{M}_\theta^{(\ell)}\Big(\mathcal{H}^{(\ell-1)}\Big) = \big\{\,\mathbf{h}_1^{(\ell)}, \ldots, \mathbf{h}_T^{(\ell)}\,\big\}, \quad \ell = 1, \ldots, L. \tag{2}$$

To distill the fine-grained reasoning result, let $t_i^*$ denote the position index of the $i$-th attribute-specific query token $\texttt{<SEG>}_i$ within the sequence. We extract the last-layer hidden state at this specific position to obtain the final reasoned embedding $\mathbf{h}_{t_i^*}^{(L)}$. Aggregating these embeddings over all $N$ attributes yields the reasoning set $\mathbf{H}_{\text{reason}} = \{\mathbf{h}_{t_i^*}^{(L)}\}_{i=1}^N$, which provides disentangled, spatially-aware guidance for the segmentation decoder.

**LoRA-based Adaptation.** To efficiently adapt the pretrained $\mathcal{M}_\theta$ to CTV segmentation, we employ Low-Rank Adaptation (LoRA) (Hu et al., 2022) on the query and value projections. Instead of full-parameter updates, the weight transformation is parameterized as $W' = W + BA$, where $A \in \mathbb{R}^{r \times d'}$ and $B \in \mathbb{R}^{d' \times r}$ represent learnable low-rank matrices ($r \ll d'$). This strategy minimizes computational overhead while enabling the model to effectively capture the attribute correspondences essential for CTV segmentation.

### 2.3. Training Objective

We adopt a weighted combination of Dice loss and binary cross-entropy (BCE) to optimize the network, formulated as:

$$\mathcal{L}_{\text{total}} = \lambda_0 \, \mathcal{L}_{\text{Dice}}(\hat{\mathbf{y}}_i, \mathbf{y}_i) + \lambda_1 \, \mathcal{L}_{\text{BCE}}(\hat{\mathbf{y}}_i, \mathbf{y}_i), \tag{3}$$

where $\hat{\mathbf{y}}_i$ and $\mathbf{y}_i$ denote the predicted and ground-truth CTV masks, respectively, and $\lambda_0$, $\lambda_1$ are weighting coefficients. The Dice and BCE loss are formulated as $\mathcal{L}_{\text{Dice}} = 1 - \dfrac{2\sum_i \hat{\mathbf{y}}_i \mathbf{y}_i + \epsilon}{\sum_i \hat{\mathbf{y}}_i + \sum_i \mathbf{y}_i + \epsilon}$ and $\mathcal{L}_{\text{BCE}} = -\dfrac{1}{|\Omega|} \sum_{j \in \Omega} [\mathbf{y}_j \log \hat{\mathbf{y}}_j + (1 - \mathbf{y}_j) \log(1 - \hat{\mathbf{y}}_j)]$, where $|\Omega|$ is the number of voxels and $\epsilon$ is a small constant for numerical stability.

## 3. Experiments

### 3.1. Datasets and Implementation Details

We conduct experiments on a large-scale multimodal in-house dataset comprising 688 post-operative prostate cancer patients, collected from the Department of Radiation Oncology at UT Southwestern Medical Center. This cohort represents a clinically demanding scenario where the prostate has been surgically removed, requiring the CTV to be inferred by synthesizing surrounding anatomical landmarks (e.g., bladder, rectal wall) with patient-specific pathological attributes. Ground-truth CTV masks were manually delineated by six experienced radiation oncologists following consensus guidelines. In addition, the operative pathology report associated with each patient includes critical pathological attributes, such as pathological T-stage, Gleason Score, and Seminal Vesicle Invasion (SVI) status. To ensure rigorous evaluation, we perform five randomized training-validation splits. Specifically, in each iteration, approximately 90% ($N_{\text{train}} = 496$) of the data is allocated for training and 10% ($N_{\text{val}} = 54$) for validation. Furthermore, 138 cases are reserved as a fixed hold-out test set to assess the final performance. All data splitting is performed at the patient level to prevent data leakage.

All CT volumes are preprocessed following the nnU-Net (Isensee et al., 2021) pipeline, including isotropic resampling to $1 \times 1 \times 1$ mm$^3$, intensity normalization, and foreground cropping. The model is implemented in MONAI (Cardoso et al., 2022) with a patch size of $320 \times 320 \times 64$ and batch size 1. The M3D-ViT input resolution is $32 \times 256 \times 256$, consistent with its pre-training setup, and the maximum LLM context length is 512 with 128 visual tokens. LoRA fine-tuning is applied to the query and value projections ($q\_proj, v\_proj$) using rank $r = 16$, scaling factor $\alpha = 16$, and dropout 0.05. Standard data augmentation (e.g., rotation, scaling, flipping) is used during training. The network is optimized with AdamW (Loshchilov and Hutter, 2019) (lr=$1 \times 10^{-5}$, weight decay=$1 \times 10^{-8}$) for up to

50 epochs on a cluster equipped with six NVIDIA H100 GPUs. Performance is evaluated using Dice Similarity Coefficient (Dice), 95th Percentile Hausdorff Distance (HD95), and Average Symmetric Surface Distance (ASSD).

### 3.2. Comparative Results

To evaluate the segmentation accuracy of ReaCT, we compare it with both vision-only and multimodal segmentation baselines. As summarized in Table 1, ReaCT achieves state-of-the-art performance, with a Dice score of 0.8185, HD95 of 4.15 mm, and ASSD of 1.38 mm. First, vision-only baselines (e.g., 3D U-Net, nnU-Net) achieve relatively stable Dice results; however, the HD95 and ASSD metrics still indicate a deficiency in identifying precise CTV boundaries. This highlights the importance of pathological information to provide critical guidance for regions that are radiographically indistinguishable based on imaging modalities alone.

Second, to justify the computational cost of employing a large LLM backbone, we compare ReaCT against two simple and resource-efficient metadata fusion baselines: (1) a 3D U-Net with one-hot encoded clinical attributes, where each attribute is represented as an orthogonal binary vector and (2) a 3D U-Net with concatenated text embeddings, where pathological attributes are encoded using a pretrained biomedical text encoder PubMed-BERT. For both baselines, we follow the fusion strategy of the CLIP-Driven Universal Model (Liu et al., 2023), concatenating the attribute representations with global visual features obtained via global average pooling applied to the final encoder layer. As shown in Table 1, simple text embedding concatenation leads to a performance drop compared to the vision-only baseline, as directly fusing high-dimensional textual representations with visual features introduces a substantial modality gap. While one-hot encoding yields only a marginal improvement (+0.34% Dice), this representation treats clinical attributes as mathematically independent symbols with zero similarity, making it difficult to capture the structured and correlated dependencies implied by clinical guidelines (e.g., Stage T3b inherently implies seminal vesicle invasion). In contrast, ReaCT substantially outperforms both one-hot encoding (+3.04% Dice) and text embedding concatenation (+5.1% Dice), demonstrating that the performance gains stem from explicit multimodal reasoning rather than simple feature augmentation. Third, text-prompted segmentation methods, including BiomedParse (Zhao et al., 2025a), SAT (Zhao et al., 2025b), and Medformer (Rajendran et al., 2025), show worse performance than vision-only baselines. We attribute this to their reliance on text encoders with encoder-only architectures. While these text encoders excel at processing explicit, content-descriptive prompts, they lack the reasoning capacity required to translate abstract and implicit pathological attributes into effective segmentation embeddings. Consequently, while their fixed prompt templates allow for flexible text-conditioned segmentation, they fall short in the deep clinical reasoning essential for CTV segmentation. Furthermore, compared to LLMSeg, which similarly employs LLaMA-2 to combine electronic medical records with images, ReaCT incorporates a more comprehensive set of attributes to enable fine-grained reasoning in postoperative scenarios where the GTV has been surgically removed. Moreover, ReaCT adopts an MLLM reasoner that jointly processes visual and textual tokens, which promotes more effective reasoning than relying on textual information only. Qualitative visualizations provided in the Appendix B.3.2 further

Table 1: Quantitative results of CTV segmentation on the in-house dataset (Mean ± SD). ↑: higher is better; ↓: lower is better.

| Methods | Dice ↑ | HD95 (mm) ↓ | ASSD (mm) ↓ |
|---|---|---|---|
| 3D U-Net (Çiçek et al., 2016) | $0.7847_{\pm 0.01}$ | $6.97_{\pm 2.33}$ | $2.13_{\pm 0.54}$ |
| nnU-Net (Isensee et al., 2021) | $0.7822_{\pm 0.01}$ | $11.69_{\pm 4.31}$ | $3.70_{\pm 0.85}$ |
| UNETR (Hatamizadeh et al., 2022) | $0.7843_{\pm 0.01}$ | $7.02_{\pm 2.20}$ | $2.15_{\pm 0.55}$ |
| Swin-UNETR (Hatamizadeh et al., 2021) | $0.7965_{\pm 0.01}$ | $5.51_{\pm 1.25}$ | $1.88_{\pm 0.44}$ |
| U-Mamba (Ma et al., 2024) | $0.7715_{\pm 0.02}$ | $7.50_{\pm 1.80}$ | $2.25_{\pm 0.55}$ |
| BiomedParse (Zhao et al., 2025a) | $0.7680_{\pm 0.04}$ | $8.85_{\pm 1.50}$ | $2.45_{\pm 0.45}$ |
| SAT (Zhao et al., 2025b) | $0.7560_{\pm 0.03}$ | $9.20_{\pm 2.20}$ | $2.68_{\pm 0.70}$ |
| Medformer (Rajendran et al., 2025) | $0.7750_{\pm 0.06}$ | $9.80_{\pm 3.10}$ | $2.80_{\pm 0.85}$ |
| LLMSeg (Oh et al., 2024) | $0.7857_{\pm 0.01}$ | $5.20_{\pm 1.02}$ | $1.75_{\pm 0.44}$ |
| Text-Concat | $0.7675_{\pm 0.02}$ | $7.85_{\pm 2.10}$ | $2.45_{\pm 0.65}$ |
| One-Hot | $0.7881_{\pm 0.02}$ | $6.84_{\pm 1.95}$ | $2.03_{\pm 0.45}$ |
| w/o Textual Tokens | $0.8015_{\pm 0.05}$ | $4.52_{\pm 0.75}$ | $1.51_{\pm 0.18}$ |
| w/o Visual Tokens | $0.7942_{\pm 0.04}$ | $5.05_{\pm 0.85}$ | $1.64_{\pm 0.20}$ |
| w/o MLLM Reasoner | $0.7885_{\pm 0.02}$ | $5.85_{\pm 1.10}$ | $1.92_{\pm 0.40}$ |
| Generic Text Prompt | $0.7989_{\pm 0.04}$ | $4.98_{\pm 0.80}$ | $1.63_{\pm 0.19}$ |
| Concat Attributes | $0.8017_{\pm 0.05}$ | $4.65_{\pm 0.70}$ | $1.56_{\pm 0.15}$ |
| **ReaCT (Ours)** | $\mathbf{0.8185^{**}_{\pm 0.05}}$ | $\mathbf{4.15^{***}_{\pm 1.66}}$ | $\mathbf{1.38^{***}_{\pm 0.48}}$ |

[**] and [***] indicate statistically significant improvements over the strongest ablation baseline (*w/o Textual Tokens*) based on the Wilcoxon signed-rank test. Specifically, [**] denotes $p < 0.01$ and [***] denotes $p < 0.001$.

demonstrate that ReaCT produces contours with better anatomical consistency, especially in challenging regions where boundaries are ambiguous.

### 3.3. Ablation Results

3.3.1. IMPACT OF MULTIMODAL REASONING COMPONENTS.

We conduct ablation studies to validate the contributions of the MLLM-based reasoner and the specific prompt design strategies. First, to examine the role of each modality, we evaluate variants using only image tokens (*w/o Textual Tokens*) or only attribute text (*w/o Visual Tokens*) as input to the MLLM. In the *w/o Textual Tokens* setting, all attribute tokens are removed from the input sequence. Following the design of M3D-LaMed (Bai et al., 2024), the input to the LLaMA backbone consists solely of visual tokens extracted by the M3D-ViT image encoder, followed by the 3D spatial pooling projector. The LLaMA backbone processes these visual tokens using causal self-attention, modeling global semantic relationships among visual features without relying on any textual queries. As shown in Table 1, both variants exhibit clear performance drops compared to ReaCT, indicating that neither modality alone is sufficient for accurate CTV segmentation. Notably, the *w/o Textual Tokens* variant (Dice: 0.8015) outperforms standard vision-only baselines such as nnU-Net. We attribute this improvement not to parameter capacity alone, but to semantic priors transferred from large-scale language pretraining. As demonstrated in recent

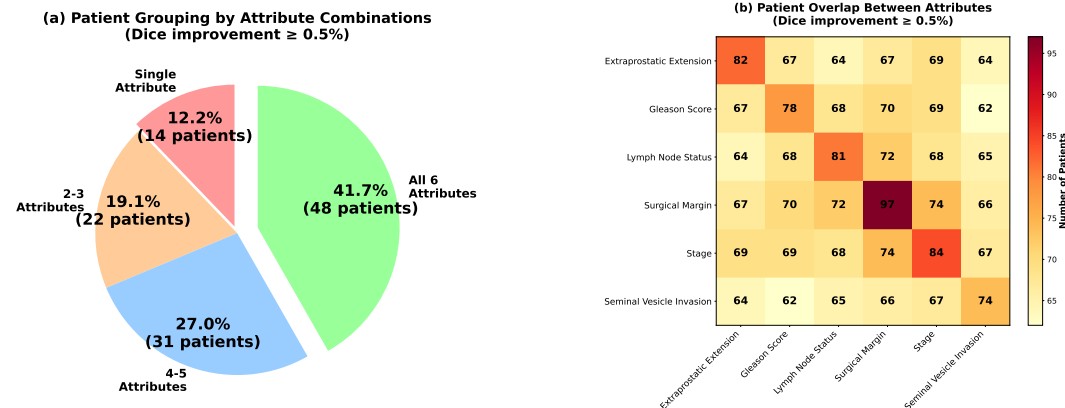

Figure 2: Analysis of Clinical Attribute Efficacy. (a) shows that most cases benefit from multi-attribute reasoning, while (b) reveals the biological correlations and complementary nature of these pathological attributes.

research (Tang et al., 2025), frozen LLM layers effectively function as semantic-aware visual boosters, where linguistic priors significantly enhance global visual representations even when processing visual tokens only. However, textual attributes remain essential for attribute-conditioned spatial reasoning. The *w/o Visual Tokens* variant (Dice: 0.7942) still outperforms standard text-conditioned baselines such as BiomedParse (Zhao et al., 2025a) and SAT (Zhao et al., 2025b), indicating that the autoregressive MLLM architecture is more effective at modeling abstract and implicit pathological attributes compared to conventional encoder-only text embeddings (e.g., CLIP or PubMedBERT). Furthermore, as shown in Appendix B.3.3, corrupting individual pathological attributes causes systematic, attribute-dependent performance degradation, confirming explicit reliance on attribute-conditioned reasoning rather than generic model capacity. Furthermore, replacing the MLLM reasoner with a standard biomedical text encoder (PubMedBERT (Gu et al., 2021)) followed by Two-Way Transformer fusion (*w/o MLLM Reasoner*) leads to a substantial degradation. This demonstrates that encoder-only text encoders are insufficient for capturing the conditional reasoning required for this task, whereas the MLLM provides essential joint reasoning capabilities.

### 3.3.2. Effectiveness of Fine-Grained Attribute Reasoning

To verify the necessity of our fine-grained, multi-token design, we compare ReaCT against two alternative prompting strategies: (1) Generic Text Prompt, which uses a static instruction (i.e., *Segment the postoperative Clinical Target Volume for prostate cancer based on the CT image*); and (2) Concat Attributes, which concatenates all extracted attributes into a single sequence followed by a single unified `<SEG>` query token. The Generic Text Prompt yields a Dice score of 0.7989, which indicates that without patient-specific context, generic instructions fail to provide effective guidance for CTV segmentation. The Concat Attributes strategy improves performance to 0.8017, yet it still lags significantly behind ReaCT. This result supports our hypothesis that compressing diverse pathological

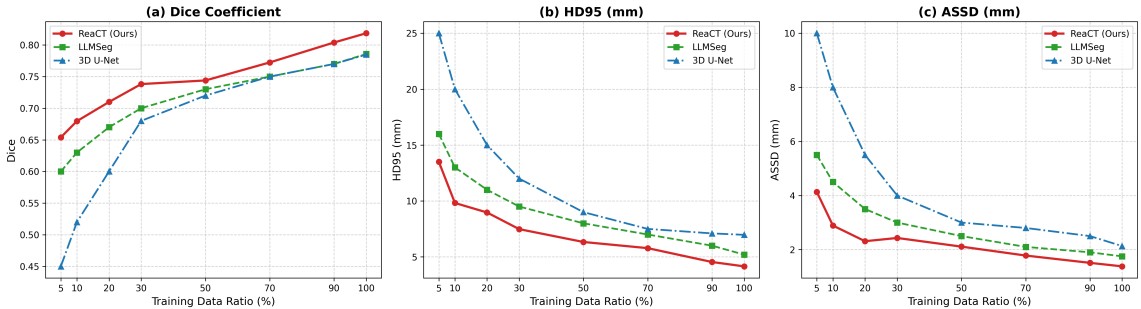

Figure 3: Performance comparison of ReaCT, 3D U-Net, and LLMSeg across varying training data ratios (5%–100%). ReaCT consistently demonstrates superior performance even in severe low-data regimes.

factors into a single global representation creates a semantic bottleneck, preventing the model from disentangling their distinct spatial implications. In contrast, ReaCT's use of a sequence of attribute-specific query tokens enables the model to explicitly reason about how each attribute dictates local boundaries, leading to improved segmentation accuracy. Appendix B.3.3 further confirms this reliance on active semantic reasoning, where deliberately corrupting attribute values leads to significant performance degradation.

### 3.3.3. Impact of Individual and Combinatorial Attributes.

To further validate the necessity of each extracted attribute, we analyze the patient cohort where ReaCT yields significant segmentation improvements (i.e. $\Delta$Dice $\geq 0.5\%$). As shown in Figure 2(a), the majority of patients (87.8%) benefit from the integration of multiple clinical attributes rather than single ones, among the six pathological attributes used in this study. This highlights the complementary nature of clinical information and the heterogeneity of patient-specific treatment responses. Furthermore, the attribute overlap matrix in Figure 2(b) reveals strong inter-attribute correlations, suggesting that different pathological factors capture overlapping yet distinct aspects of tumor characteristics. For instance, high Gleason scores frequently co-occur with positive lymph node status, reflecting the known biological propensity for high-grade tumors to metastasize. Notably, Surgical Margin, Pathological Stage, and Extraprostatic Extension emerge as the most influential factors, showing the highest diagonal densities. This is consistent with clinical practice, where these parameters serve as key determinants in defining CTV expansion boundaries.

### 3.3.4. Data Efficiency and Robustness Analysis

To investigate sample efficiency, we evaluate the performance of ReaCT against a representative vision-only baseline (3D U-Net) and a multimodal baseline (LLMSeg) under varying training data proportions ranging from 5% to 100%. As illustrated in Figure 3, ReaCT consistently outperforms both baselines across all data regimes. Notably, the 3D U-Net suffers severe degradation in extreme low-data settings (e.g., 5%–20%), as indicated by a sharp

drop in metrics. This confirms that without semantic guidance, visual features alone are insufficient to generalize from sparse supervision. While LLMSeg exhibits better stability than the vision-only model, it still lags behind ReaCT. These results highlight that explicitly modeling pathological reasoning substantially improves label efficiency and robustness, even when pixel-level supervision is scarce.

### 3.4. Discussion

While ReaCT achieves strong performance by reformulating postoperative CTV segmentation as a multimodal reasoning task, we acknowledge that this study is conducted on a single-center in-house dataset, reflecting the limited availability of large-scale public benchmarks for postoperative prostate CTV segmentation that include paired pathology reports. Importantly, ReaCT is designed to reduce institution-specific bias by conditioning segmentation on explicit pathological attributes and consensus guideline-driven rules, rather than implicitly learning local contouring styles from image appearance alone. Unlike vision-only methods that may overfit site-dependent practices, ReaCT focuses on pathology-dependent spatial decisions (e.g., margin status) that are defined by widely adopted clinical guidelines and shared across institutions. As a result, although the current evaluation is single-center, the learned reasoning is conceptually decoupled from local annotation habits and is intended to generalize once corresponding multimodal inputs are available. Moreover, clinical guidelines for postoperative radiotherapy evolve over time and may vary across institutions. ReaCT explicitly separates guideline interpretation from the segmentation network by encapsulating clinical knowledge within the LLM-based attribute extraction module, while the segmentation model operates on structured attributes and visual evidence. This modular design enables adaptation to updated or institution-specific guidelines by revising attribute definitions or prompt schemas without retraining the segmentation model itself. Such separation mirrors real-world radiotherapy workflows and enhances the transparency, controllability, and long-term clinical applicability of the proposed framework.

### 4. Conclusion

In this work, we present **ReaCT**, a unified framework reformulating CTV segmentation as a multimodal reasoning task. By designing a Guideline-Informed Attribute Extractor to distill determinative attributes and an Attribute-Specific MLLM Reasoner for fine-grained spatial inference, ReaCT effectively bridges the gap between abstract clinical logic and invisible anatomical boundaries. Experiments on a large-scale prostate dataset demonstrate state-of-the-art performance and remarkable robustness in limited-annotation regimes. Future work will focus on adaptively integrating evolving clinical guidelines to enhance generalization across diverse disease sites and institutional standards.

### Acknowledgments

This work was partially supported by US National Science Foundation IIS-2412195, CCF-2400785, the Cancer Prevention and Research Institute of Texas (CPRIT) award (RP230363), the National Institutes of Health (NIH) R01 award (1R01AI190103-01) and Microsoft Accelerate Foundation Models Research (2024).

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

## Appendix A. Related Work

### A.1. CTV Segmentation Methods

Compared to conventional segmentation tasks driven by visual contrast, Clinical Target Volume (CTV) segmentation requires identifying microscopic spread that is invisible on standard imaging. To bridge this visibility gap, previous methods have attempted to infer target boundaries using auxiliary geometric cues or handcrafted anatomical heuristics. For example, Cardenas *et al.* (Cardenas et al., 2018) proposed a dual-channel 3D U-Net that ingests both CT scans and Gross Tumor Volume (GTV) masks to infer target boundaries based on spatial proximity. Similarly, Jin *et al.* (Jin et al., 2021) introduced a framework incorporating signed distance maps from the GTV and adjacent organs to provide explicit geometric constraints. Specific to postoperative prostate cancer, Wang *et al.* (Wang et al., 2022) modeled the prostate bed as a virtual target to guide segmentation in the absence of the primary organ. While these approaches improve consistency, their reliance on fixed geometric heuristics or spatial expansions limits their adaptability to anatomical variations, particularly in postoperative scenarios where the tumor has been surgically removed. In contrast, ReaCT addresses the intrinsic need for multimodal integration in CTV segmentation, enabling clinically informed reasoning about target extent that aligns with the decision-making process of radiation oncologists.

### A.2. LLM-based CTV Segmentation Methods

The integration of Large Language Models (LLMs) into radiotherapy workflows marks a significant shift towards utilizing clinical data as auxiliary information for target volume segmentation. For instance, LLMSeg (Oh et al., 2024) demonstrated the capability of LLMs to enhance CTV segmentation by encoding clinical texts, such as tumor stage and surgery type, for breast and prostate cancer. Building on this, RO-LMM (Kim et al., 2025) proposed a comprehensive agent covering tasks from report summarization to plan-guided segmentation, while Medformer (Rajendran et al., 2024, 2025) leveraged hierarchical vision transformers fused with LLM-extracted text features to improve target delineation. However, these existing methods largely treat LLMs as static text encoders that offer only coarse global conditioning, without exploiting their reasoning capacity to model how individual pathological factors influence local anatomical boundaries. Consequently, they fail to deliver the fine-grained, attribute-specific reasoning required for accurate CTV segmentation. In contrast, ReaCT introduces a guideline-informed attribute schema and an attribute-specific multimodal LLM that performs fine-grained reasoning over visual and textual cues, enabling clinically coherent and anatomically precise boundary prediction.

## Appendix B. Details of the LLM-based Attribute Extractor

In this section, we provide detailed prompt designs and workflow specifications for the LLM-based Attribute Extractor. The module is implemented using GPT-4o (Hurst et al., 2024) and follows a multi-stage, schema-constrained pipeline designed to extract structured pathological attributes from free-text clinical reports in accordance with clinical consensus guidelines. The overall procedure is summarized in Algorithm 1, which provides an explicit description of the full extraction workflow.

---

**Algorithm 1** LLM-based Attribute Extractor Pipeline

---

**Input**              : Raw Pathology Report $D_{raw}$, Consensus Clinical Guidelines $\mathcal{G}$
**Output**             : Structured Clinical Attribute Set $\mathcal{A}$
**Hyperparameters:** Schema Prompt $\mathcal{P}_{schema}$, Retrieval Prompt $\mathcal{P}_{retrieval}$, Extraction Prompt
                     $\mathcal{P}_{extract}$

```
// Stage 1:  Guideline Knowledge Distillation (Appendix B.1)
```
$\mathcal{S} \leftarrow \mathrm{LLM}(\mathcal{P}_{schema}, \mathcal{G})$ ;                    `// Distill determinative attribute schema`
**Freeze** schema $\mathcal{S}$ for inference

```
// Stage 2:  Patient-Specific Inference (Appendix B.2)
```
$\mathcal{A} \leftarrow \emptyset$ **for** *each patient report $D_{raw}$* **do**
  | `// Step 2.1:  Relevant Context Retrieval`
  | $T_{rel} \leftarrow \mathrm{LLM}(\mathcal{P}_{retrieval}, \mathcal{S}, D_{raw})$ ;                    `// Filter irrelevant history`
  | `// Step 2.2:  Semantic Verification & Standardization`
  | $V_{raw} \leftarrow \mathrm{LLM}(\mathcal{P}_{extract}, T_{rel})$ **for** *each attribute $k \in \mathcal{S}$* **do**
  |   | $v_k \leftarrow \mathrm{Standardize}(V_{raw}[k])$ ;                    `// Map to standard values`
  |   | $\mathcal{A}.\mathrm{append}((k, v_k))$
  | **end**
**end**
**return** $\mathcal{A}$

---

## B.1. Guideline-Based Schema Construction

The objective of this stage is to distill a fixed, principled schema of determinative attributes from authoritative sources. We first aggregate relevant clinical guidelines (e.g., ESTRO ACROP, NCCN, RTOG) retrieved from medical databases such as PubMed. Based on this compiled corpus, we employ a Knowledge Distillation Prompt $\mathcal{P}_{\text{schema}}(\cdot)$ that instructs the LLM to act as a domain expert to synthesize a standardized attribute list. The prompt is specifically designed to identify pathological factors that dictate boundary modifications for CTV segmentation, consolidating diverse guideline terminologies into a unified schema.

---

**Prompt 1: Guideline Knowledge Distillation**

**System Role:** You are a board-certified radiation oncologist and expert in prostate cancer radiotherapy planning.

**Context:** Accurate delineation of the Clinical Target Volume (CTV) for postoperative prostate cancer relies on specific pathological risk factors defined in consensus guidelines.

**Task:** Read the aggregated guideline documents provided below. Identify and summarize the specific pathological attributes that explicitly govern the anatomical boundaries of the CTV. For each attribute, explain how it influences the target volume (e.g., "inclusion of seminal vesicle bed").

**Input Guidelines:** [Insert full text of compiled ESTRO / NCCN / RTOG guidelines here...]

**Requirements:**

1. Output a structured list of determinative attributes (e.g., T-Stage, Gleason Score).
2. Focus strictly on factors influencing anatomical target boundaries.
3. Consolidate synonymous terms into a standardized schema key.

**Output Format:** JSON list of keys.

---

Based on the output, we finalized the attribute schema $\mathcal{S}$ by retaining factors with explicit spatial implications for CTV delineation. This selection was further verified by senior radiation oncologists to ensure alignment with clinical consensus. The six determinative attributes are: **Pathological T-Stage**, **Gleason Score**, **Seminal Vesicle Invasion**, **Extraprostatic Extension**, **Surgical Margin Status**, and **Lymph Node Status**.

### B.2. Multi-Stage Attribute Extraction

This stage transforms lengthy, unstructured pathology reports into the structured attribute profile $\mathcal{A}$. The process involves a context retrieval step followed by semantic verification and value standardization.

**1. Relevant Context Retrieval:** To efficiently narrow the search space within lengthy patient records, we employ a Context Retrieval Prompt that functions as the operator $\mathcal{F}_{\text{key}}(\cdot)$. This step filters the raw document $\mathbf{D}_{\text{raw}}$ to identify candidate text spans related to the schema $\mathcal{S}$, strictly excluding irrelevant medical history. The output list constitutes the **relevant text set $\mathbf{T}_{\text{rel}}$**, which serves as the input for the subsequent verification step.

---

**Prompt 2: Relevant Context Retrieval**

**System Role:** You are assisting in extracting pathological attributes for postoperative prostate cancer.
**Task:** Given the schema below, identify all sentences or short text spans from the pathology report that may contain information relevant to any attribute in the schema.
**Schema:** [Insert Attribute Schema $\mathcal{S}$ derived from Prompt 1]
**Pathology Report:** [Insert Raw Pathology Report $\mathbf{D}_{\text{raw}}$]
**Output:**

1. A list of relevant text spans (verbatim from the report).
2. Do NOT infer values yet; only retrieve candidate segments.

---

**2. Semantic Verification & Standardization:** We design a clinical extraction prompt to process the retrieved context $\mathbf{T}_{\text{rel}}$. This prompt is designed to perform semantic verification (e.g., distinguishing "margins are negative" from "margins were not assessed") and to standardize attribute values into a structured form suitable for downstream multimodal reasoning.

---

**Prompt 3: Attribute Verification and Standardization**

**System Role:** You are an expert pathologist. Your task is to extract structured clinical variables from the provided text segments of a radical prostatectomy pathology report.
**Input Text:** [Insert filtered text segments $\mathbf{T}_{\text{rel}}$ from Prompt 2]
**Target Schema:** Extract values for the following attributes:

1. Pathological T-Stage
2. Gleason Score (e.g., 7(3+4))
3. Seminal Vesicle Invasion (SVI)
4. Extraprostatic Extension (EPE)
5. Surgical Margin Status
6. Lymph Node Status

**Instructions:**

1. **Semantic Verification:** Ignore text related to previous biopsy history or other irrelevant procedures. Focus only on the final surgical pathology.
2. **Redundancy Removal:** If multiple mentions exist, prioritize the "Final Diagnosis" section.
3. **Standardization:** Map the extracted values to the following standard formats:

    - SVI/EPE/Margins/Nodes: "Positive" or "Negative".
    - T-Stage: e.g., "pT2", "pT3a", "pT3b".
    - If an attribute is not mentioned or cannot be determined, output "Unknown".

**Output Format:** Provide the result as a JSON object: {"Attribute": "Standardized Value"}.

---

## B.3. Additional Experiments

### B.3.1. VALIDATION OF LLM-BASED ATTRIBUTE EXTRACTION

To assess the reliability of GPT-4o for extracting clinical attributes from pathology reports, we conducted a quantitative validation study. We randomly sampled 200 cases from our dataset and asked two radiation oncologists with experience in postoperative prostate radiotherapy to independently verify the extracted attributes against the original pathology reports. In cases of disagreement, a consensus annotation was reached through joint discussion, and the final consensus labels were used as ground truth for evaluation. We report both accuracy and F1-score for each attribute, as well as overall performance across all attributes. Table 2 summarizes the extraction performance for the six clinical attributes. The overall extraction accuracy was 97.1% with an F1-score of 0.96, demonstrating the high reliability of the automated extraction pipeline. These results indicate that the guideline-informed extraction achieves high reliability and provides a stable foundation for downstream multimodal segmentation.

### B.3.2. QUALITATIVE COMPARISON

Figure 4 presents a qualitative comparison of segmentation results across five representative patients from the test set. To rigorously assess clinical plausibility and boundary behavior, we deliberately selected cases spanning diverse pathological conditions, including varying Gleason scores (ranging from 3+4 to 4+5) and different extents and locations

Table 2: GPT-4o attribute extraction validation results on 200 randomly sampled cases.

| Attribute | Accuracy (%) | F1-Score |
|---|---|---|
| Stage | 98.5 | 0.98 |
| Gleason Score | 97.5 | 0.97 |
| Extraprostatic Extension | 96.0 | 0.95 |
| Lymph Node Status | 98.0 | 0.97 |
| Surgical Margin | 96.5 | 0.96 |
| Seminal Vesicle Invasion | 96.0 | 0.95 |
| **Overall** | **97.1** | **0.96** |

of positive surgical margins (e.g., left apex, right postero-lateral, and multifocal invasive carcinoma). To facilitate detailed inspection of boundary differences, synchronized region-of-interest (ROI) zoom-in views are provided for all methods. As observed in these zoomed regions, conventional vision-only models (e.g., U-Net (Çiçek et al., 2016), nnU-Net (Isensee et al., 2021), and UNETR (Hatamizadeh et al., 2022)) generally capture the coarse CTV extent but consistently exhibit noticeable deviations at anatomically intricate boundaries, often producing overly smoothed or imprecise contours that fail to strictly adhere to clinical guidelines. Similarly, representative text-conditioned segmentation approaches (e.g., BiomedParse (Zhao et al., 2025a) and SAT (Zhao et al., 2025b)) do not demonstrate consistent improvements in these boundary-critical regions, indicating that direct image–text fusion via feature concatenation or cross-attention is insufficient to bridge the modality gap associated with highly implicit clinical text. In contrast, ReaCT consistently adapts its boundary behavior across diverse pathological scenarios while maintaining stable and clinically plausible contours. This qualitative evidence demonstrates that ReaCT can accommodate heterogeneous pathological presentations without sacrificing boundary accuracy, supporting its practical applicability for postoperative CTV delineation.

### B.3.3. Robustness and Sensitivity Analysis

To verify that ReaCT actively leverages clinical attributes for decision-making rather than treating text as a passive feature, we conducted a robustness analysis by deliberately corrupting individual attributes during inference. Specifically, for each experiment, we flipped the value of a single determinative attribute to its opposite clinical status (e.g., changing *Surgical Margin Status* from "Positive" to "Negative" or vice versa) while keeping all other attributes and the image input unchanged. This setup isolates the causal impact of each specific attribute on the segmentation outcome. As shown in Table 3, incorrect pathological inputs consistently degrade performance. Notably, corrupting the Surgical Margin status causes the most significant drop (Dice $-2.61\%$, HD95 $+0.67$ mm). This aligns with clinical guidelines where positive margins mandate aggressive CTV expansion, significantly altering target geometry. Similarly, incorrect Extraprostatic Extension and Pathological Stage inputs also lead to marked performance losses, confirming the model's dependency on accurate determinative factors.

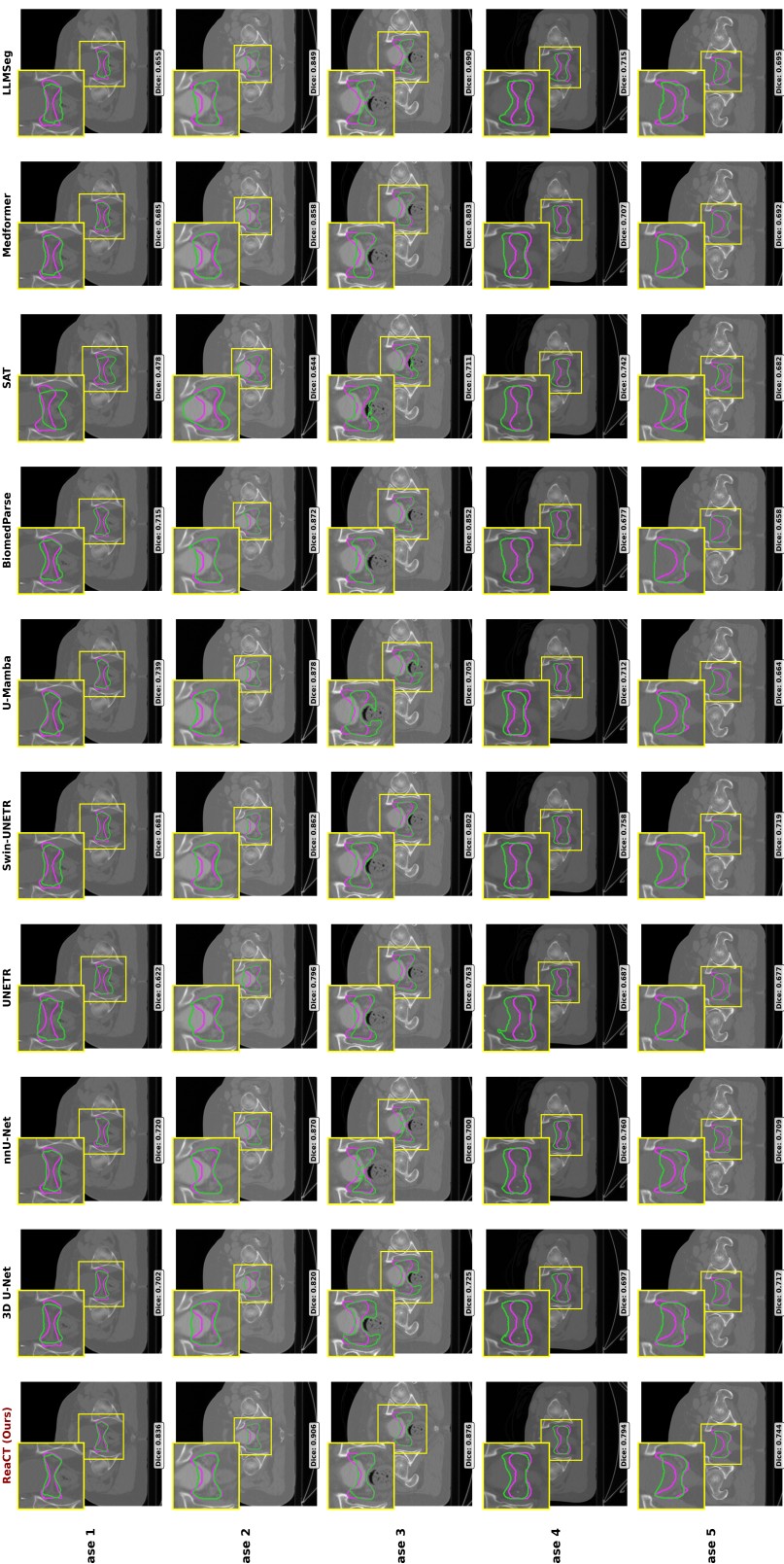

Figure 4: Qualitative comparison on five representative cases. The Ground Truth is outlined in pink, and model predictions are in green. Columns: **ReaCT (Ours)**, 3D U-Net, nnU-Net, BiomedParse, and LLMSeg. ReaCT consistently demonstrates superior alignment with the ground truth in radiographically ambiguous regions compared to baseline methods.

Table 3: Robustness Analysis. Performance degradation when individual clinical attributes are corrupted. Each row shows results when a single attribute is deliberately flipped to its opposite clinical value (e.g., Positive $\leftrightarrow$ Negative) while keeping all other attributes correct.

| Model Variant | Dice $\uparrow$ | HD95 (mm) $\downarrow$ | ASSD (mm) $\downarrow$ |
|---|---|---|---|
| **ReaCT (Original)** | $\mathbf{0.8185}_{\pm 0.05}$ | $\mathbf{4.15}_{\pm 1.66}$ | $\mathbf{1.38}_{\pm 0.48}$ |
| ReaCT w/ Wrong SM | $0.7924_{\pm 0.06}\ (-2.61\%)$ | $4.82_{\pm 1.89}\ (+0.67)$ | $1.56_{\pm 0.52}\ (+0.18)$ |
| ReaCT w/ Wrong EPE | $0.7979_{\pm 0.05}\ (-2.06\%)$ | $4.68_{\pm 1.78}\ (+0.53)$ | $1.52_{\pm 0.51}\ (+0.14)$ |
| ReaCT w/ Wrong Stage | $0.7995_{\pm 0.05}\ (-1.90\%)$ | $4.59_{\pm 1.74}\ (+0.44)$ | $1.49_{\pm 0.50}\ (+0.11)$ |
| ReaCT w/ Wrong GS | $0.8036_{\pm 0.05}\ (-1.49\%)$ | $4.42_{\pm 1.71}\ (+0.27)$ | $1.45_{\pm 0.49}\ (+0.07)$ |
| ReaCT w/ Wrong LNS | $0.8064_{\pm 0.05}\ (-1.21\%)$ | $4.35_{\pm 1.69}\ (+0.20)$ | $1.43_{\pm 0.49}\ (+0.05)$ |
| ReaCT w/ Wrong SVI | $0.8077_{\pm 0.05}\ (-1.08\%)$ | $4.31_{\pm 1.68}\ (+0.16)$ | $1.42_{\pm 0.49}\ (+0.04)$ |

Table 4: Leave-one-out attribute ablation on ReaCT. Each row removes one clinical attribute while keeping all others unchanged. Reported values show absolute performance and relative change compared to the full model.

| Model Variant | Dice $\uparrow$ | HD95 (mm) $\downarrow$ | ASSD (mm) $\downarrow$ |
|---|---|---|---|
| **ReaCT (6 attrs)** | $\mathbf{0.8185}_{\pm 0.05}$ | $\mathbf{4.15}_{\pm 1.66}$ | $\mathbf{1.38}_{\pm 0.48}$ |
| w/o Stage | $0.8092_{\pm 0.05}\ (-0.93\%)$ | $4.48_{\pm 1.71}\ (+0.33)$ | $1.47_{\pm 0.50}\ (+0.09)$ |
| w/o Gleason Score | $0.8101_{\pm 0.05}\ (-0.84\%)$ | $4.44_{\pm 1.70}\ (+0.29)$ | $1.46_{\pm 0.49}\ (+0.08)$ |
| w/o Extraprostatic Extension | $0.8048_{\pm 0.05}\ (-1.37\%)$ | $4.62_{\pm 1.76}\ (+0.47)$ | $1.51_{\pm 0.51}\ (+0.13)$ |
| w/o Seminal Vesicle Invasion | $0.8062_{\pm 0.05}\ (-1.23\%)$ | $4.55_{\pm 1.74}\ (+0.40)$ | $1.49_{\pm 0.50}\ (+0.11)$ |
| w/o Surgical Margin | $0.8019_{\pm 0.05}\ (-1.66\%)$ | $4.82_{\pm 1.89}\ (+0.67)$ | $1.56_{\pm 0.52}\ (+0.18)$ |
| w/o Lymph Node Status | $0.8085_{\pm 0.05}\ (-1.00\%)$ | $4.41_{\pm 1.69}\ (+0.26)$ | $1.45_{\pm 0.49}\ (+0.07)$ |

### B.3.4. Leave-One-Out Attribute Ablation

To further examine whether the selected pathological attributes are redundant or jointly necessary, we conduct a leave-one-out attribute ablation study on the full ReaCT model. In this analysis, each clinical attribute is removed in turn while all remaining attributes and network components are kept unchanged. As shown in Table 4, removing any single attribute consistently leads to performance degradation across all three metrics, indicating that no attribute can be omitted without measurable loss in segmentation quality. Notably, attributes that directly determine boundary expansion decisions in clinical guidelines, such as surgical margin status, extraprostatic extension, and seminal vesicle invasion, exhibit the largest performance drops when removed, reflecting their critical role in postoperative CTV delineation. Overall, this leave-one-out analysis demonstrates that the six pathological attributes are not redundant but instead provide complementary clinical information. The results support the conclusion that all selected attributes are jointly necessary for robust and guideline-consistent postoperative CTV segmentation.

Table 5: Sensitivity analysis of the Dice improvement threshold used to identify patients where integrating clinical attributes achieves meaningful segmentation gains over the baseline. For each threshold, we report the number of patients meeting the criterion and the distribution of beneficial attributes.

| Threshold | Patients | 1-Attr. (%) | 2–3 Attrs. (%) | 4–5 Attrs. (%) | All 6 (%) |
|-----------|----------|-------------|----------------|----------------|-----------|
| 0.00% | $125_{\pm 5}$ | $9.6_{\pm 1.2}$ | $16.0_{\pm 1.5}$ | $28.0_{\pm 2.0}$ | $46.4_{\pm 2.5}$ |
| 0.25% | $120_{\pm 4}$ | $11.7_{\pm 1.3}$ | $17.5_{\pm 1.6}$ | $27.5_{\pm 1.8}$ | $43.3_{\pm 2.3}$ |
| 0.50% | $115_{\pm 4}$ | $12.2_{\pm 1.4}$ | $19.1_{\pm 1.7}$ | $27.0_{\pm 1.9}$ | $41.7_{\pm 2.2}$ |
| 0.75% | $108_{\pm 5}$ | $15.7_{\pm 1.6}$ | $21.3_{\pm 1.8}$ | $25.9_{\pm 2.0}$ | $37.1_{\pm 2.4}$ |
| 1.00% | $100_{\pm 5}$ | $19.0_{\pm 1.8}$ | $24.0_{\pm 2.0}$ | $25.0_{\pm 2.1}$ | $32.0_{\pm 2.5}$ |

### B.3.5. Sensitivity Analysis of Dice Threshold Selection

In Section 3.3.3, a Dice improvement threshold of 0.5% was used to identify patients for whom ReaCT yields meaningful segmentation improvements over the vision-only baseline. This threshold was chosen to distinguish clinically relevant improvements from minor variations attributable to measurement noise introduced by sliding-window aggregation and boundary discretization during volumetric inference.

To assess the robustness of our conclusions with respect to this choice and to rule out potential selection bias, we conduct a sensitivity analysis across a range of Dice thresholds, including 0.00%, 0.25%, 0.50%, 0.75%, and 1.00%. For each threshold, we recompute the subset of patients for whom ReaCT achieves Dice improvements exceeding the threshold relative to the baseline, and analyze how many clinical attributes contribute to these gains. Table 5 summarizes the results. Although stricter thresholds naturally reduce the number of included patients, the key finding remains consistent across all settings: the majority of patients (81–90%) benefit from the integration of multiple clinical attributes (i.e., two or more attributes), rather than any single attribute alone. This observation confirms that the complementary nature of clinical information is not an artifact of threshold selection. Furthermore, the attribute overlap matrix shown in Figure 2(b) demonstrates that different attributes benefit partially distinct patient subgroups, reinforcing the necessity of incorporating all six pathological attributes. Finally, we emphasize that the main quantitative results reported in Table 1 are computed on the full test set ($N = 139$) without any threshold filtering, ensuring that the overall performance conclusions are not subject to selection bias.

### B.4. Computational Cost and Clinical Deployment

We benchmark the computational requirements of ReaCT on a single NVIDIA H100 GPU. The model contains 7.02B total parameters, with only 38.5M (0.55%) trainable via LoRA adapters. Training converges in approximately 2.75 hours (26 epochs), and inference takes ∼85 ms per case with ∼27 GB VRAM in FP32 precision. For clinical deployment, half-precision (FP16) inference reduces memory requirements to ∼14 GB, compatible with standard workstation GPUs (e.g., RTX 3090/4090). The sub-second inference latency is negligi-

Table 6: Computational specifications of ReaCT.

| Metric | Value | Note |
|---|---|---|
| Total Parameters | 7.02B | Frozen M3D backbone + LoRA + 3D U-Net |
| Trainable Parameters | 38.5M (0.55%) | LoRA + 3D U-Net |
| Training Time | ~2.75 hours | 26 epochs on single H100 GPU |
| Inference Latency | ~85 ms/case | Single forward pass |
| Inference VRAM | ~27 GB | FP32 precision |

ble compared to the typical 20–30 minute manual CTV delineation time, enabling seamless integration into existing treatment planning workflows. Table 6 summarizes the computational specifications of ReaCT.

While the peak VRAM usage of ~27 GB reflects our FP32 research configuration, clinical deployment is feasible on standard workstations. In practice, half-precision (FP16) inference reduces memory requirements to approximately 13-14 GB, fitting within widely available GPUs such as the RTX 3090/4090 with 24 GB VRAM. For hardware with stricter memory constraints, established quantization techniques (INT8/INT4) (Lin et al., 2024) can further reduce requirements to below 10 GB with minimal impact on segmentation accuracy. Furthermore, postoperative CTV delineation is an offline treatment planning task that typically requires 20–30 minutes of clinician time (Cha et al., 2021), making the ~85 ms inference latency negligible in comparison and fully compatible with existing PACS/TPS workflows.

