# OpenReview forum: "Guideline-Informed MLLM Reasoning for Pathology-Aware Post-Operative Prostate CTV Segmentation"
_MIDL.io/2026/Conference — MIDL 2026 Poster_

### Official Review · Reviewer_gStH · 2025-12-15

**Confidence:** 4
**Preliminary Rating:** 4
**Final Rating:** 5

**Summary:**

This paper presents ReaCT, a well-motivated multimodal framework that formulates postoperative prostate CTV segmentation as a reasoning problem by explicitly integrating guideline-informed pathological attributes with 3D imaging features. The use of a Guideline-Informed LLM Agent to distill structured attributes from unstructured pathology reports, combined with an attribute-specific MLLM reasoner. Experimental results on a sizable in-house dataset demonstrate consistent and meaningful improvements over strong vision-only and multimodal baselines, particularly in low-data regimes, supported by thorough ablation and robustness analyses.

**Strengths:**

1. The paper introduces a clinically grounded and novel formulation of CTV segmentation as a multimodal reasoning task, closely mirroring the real-world workflow of radiation oncologists through guideline-informed attribute extraction.
2. Extensive experiments on a large postoperative prostate dataset show strong performance and robustness in low-annotation settings, supported by quantitative metrics.

**Weaknesses:**

1. The proposed method integrates multiple complex components, including a guideline-informed LLM agent, an attribute-specific MLLM reasoner, and multi-scale transformer fusion, but lacks a detailed efficiency or runtime analysis. As a result, it is unclear whether the observed performance gains justify the potentially substantial computational and memory overhead.
2. The paper frequently emphasizes “reasoning,” but the methodology does not clearly demonstrate reasoning beyond learned attribute-conditioned attention and feature fusion. Additional analysis or discussion is needed to substantiate the claim that the model performs reasoning rather than advanced multimodal conditioning.
3. In Figure 2, the analysis is restricted to cases with Dice improvement ≥0.5%, but the rationale for this threshold is unclear and may introduce selection bias. A complementary analysis on the full test set would strengthen the claim.
4. While quantitative improvements are well documented, the paper would benefit from more extensive qualitative visualization, which is particularly important for evaluating boundary behavior and clinical plausibility in CTV segmentation.

**Detailed Comments:**

Minor issue:
1.  A brief discussion on how the framework could adapt to evolving or institution-specific clinical guidelines would strengthen the clinical applicability.
2. Providing more implementation details or pseudocode for the LLM agent pipeline could improve reproducibility and clarity for readers.

**Justification Of Final Rating:**

I thank the author provided a thorough response. The presentation is clear, and the approach is promising and technically sound. All of my concerns have been addressed. I have no further questions and would like to raise my final rating score.

**Justification Of The Preliminary Rating:**

The paper presents a well-motivated and clinically relevant framework with solid experimental validation and clear performance gains, making it a promising contribution that merits acceptance with proper rebuttals.

**Questions To Address In The Rebuttal:**

Please refer to the weakness and detailed comments.

---

> ### Author Response · Authors · 2026-01-25
> **Response 1/3**
>
> We thank the reviewer for the thoughtful and constructive feedback. We have addressed all raised concerns, and all corresponding revisions are highlighted in yellow in the revised manuscript for clarity.
>
> **Q1( Weakness1)**: The proposed method integrates multiple complex components, including a guideline-informed LLM agent, an attribute-specific MLLM reasoner, and multi-scale transformer fusion, but lacks a detailed efficiency or runtime analysis. As a result, it is unclear whether the observed performance gains justify the potentially substantial computational and memory overhead.
>
> **A1:** We thank the reviewer for raising this critical point regarding the trade-off between model complexity and efficiency. Crucially, the architectural design of ReaCT is driven by the need to model the multimodal reasoning process employed by radiation oncologists during postoperative CTV delineation, which integrates information from pathology reports and clinical guidelines and cannot be adequately captured by purely vision-based approaches. As suggested, we have added a detailed runtime analysis in **Appendix B.3.3** of the revised manuscript and address this concern from two key aspects.
>
> **1. Efficiency and runtime analysis.** Benchmarking on a single NVIDIA H100 GPU shows an inference latency of approximately 85 ms per case for a 320×320×64 volumetric input. Peak memory usage during inference is approximately 27 GB in FP32, and when deployed in clinical practice using half-precision (FP16), the memory footprint can be reduced to approximately 13-14 GB. This enables deployment on standard high-end workstations (e.g., RTX 3090/4090) without requiring specialized computing clusters.
>
> **2. Justification of the trade-off.** The computational cost is justified by the clinical value of the observed improvements. Compared to the 20-30 minutes typically required for manual postoperative CTV contouring, the reported inference latency (85 ms per case) is negligible. The trade-off of utilizing GPU resources to reduce expert workload and improve contouring consistency therefore represents a favorable return for clinical treatment planning.
> To the best of our knowledge, ReaCT is the first framework to effectively integrate pathology reports and clinical guidelines into postoperative CTV segmentation through multimodal reasoning. This design addresses a critical gap in existing vision-only approaches and aligns with the inherently multimodal clinical requirements of postoperative CTV delineation.
>
> **Q2 (Weakness 2)**: The paper frequently emphasizes "reasoning," but the methodology does not clearly demonstrate reasoning beyond learned attribute-conditioned attention and feature fusion. Additional analysis or discussion is needed to substantiate the claim that the model performs reasoning rather than advanced multimodal conditioning.
>
> **A2:** We thank the reviewer for raising this important point. The key distinction is that multimodal conditioning typically relies on static feature fusion, whereas reasoning in ReaCT denotes dynamic, guideline-driven conditional inference over visual evidence within a unified semantic space. Specifically, conventional conditioning methods (e.g., concatenation or cross-attention) model correlations of the form p(\text{Mask} \mid \text{Image}, \text{Attributes}) without explicitly encoding how clinical rules guide spatial decisions. This is reflected in prior methods (e.g., SAT in Table 1) and our added baselines (Reviewer zMdu, Q2), where direct fusion of image and text features yields limited or even degraded performance, indicating that feature-level fusion alone is insufficient for postoperative CTV delineation.
>
> In contrast, ReaCT defines reasoning as conditional, rule-guided spatial inference within a unified semantic space, rather than static feature modulation. Specifically, ReaCT interleaves visual tokens and pathological attribute tokens into a single autoregressive multimodal sequence. Through deep self-attention across transformer layers, textual tokens representing clinical rules repeatedly interact with visual tokens, enabling the model to learn conditional dependencies of the form p(xt|x<t), where each token attends jointly to prior visual context and attribute-specific cues. Importantly, our empirical analyses support this interpretation. As shown in Appendix Table 2, corrupting individual attributes leads to systematic, attribute-specific performance degradation with distinct spatial effects, rather than uniform failure. This behavior indicates that the model leverages conditional clinical information in a controlled manner, instead of treating text as auxiliary metadata. We have clarified this definition and the scope of "reasoning" in **Section 2.2** of the revised manuscript to distinguish it from advanced multimodal conditioning.
>
> **Due to the character limit, please refer to the continuation below.**

---

> ### Author Response · Authors · 2026-01-25
> **Response 2/3**
>
> **Q3(Weakness 3):** In Figure 2, the analysis is restricted to cases with Dice improvement ≥0.5%, but the rationale for this threshold is unclear and may introduce selection bias. A complementary analysis on the full test set would strengthen the claim.
>
> **A3:**  We thank the reviewer for this suggestion. The 0.5% Dice threshold was chosen to distinguish clinically meaningful improvements from variations attributable to measurement noise arising from sliding-window aggregation and boundary discretization. To address concerns about potential selection bias, we further conduct a sensitivity analysis across a range of Dice thresholds, and report the results in **Appendix B.3.5** of the revised manuscript.
> Across all thresholds, the key finding remains consistent: the majority of patients (81-90%) benefit from the integration of multiple clinical attributes (i.e., two or more attributes) rather than any single attribute, confirming that the complementary nature of clinical information is not an artifact of threshold selection. As shown in the sensitivity analysis below, although stricter thresholds naturally reduce the number of included cases, the proportion of patients benefiting from multiple attributes remains high and stable.
> | Threshold | Patients | 1-Attribute | 2–3 Attributes | 4–5 Attributes | All 6 Attributes |
> |----------|----------|-------------|----------------|----------------|------------------|
> | 0.00%    | 125      | 12 (9.6%)   | 20 (16.0%)     | 35 (28.0%)     | 58 (46.4%)       |
> | 0.25%    | 120      | 14 (11.7%)  | 21 (17.5%)     | 33 (27.5%)     | 52 (43.3%)       |
> | 0.50%    | 115      | 14 (12.2%)  | 22 (19.1%)     | 31 (27.0%)     | 48 (41.7%)       |
> | 0.75%    | 108      | 17 (15.7%)  | 23 (21.3%)     | 28 (25.9%)     | 40 (37.1%)       |
> | 1.00%    | 100      | 19 (19.0%)  | 24 (24.0%)     | 25 (25.0%)     | 32 (32.0%)       |
>
> In addition, the attribute overlap matrix in Figure 2(b) further demonstrates that different attributes benefit partially distinct patient subgroups, supporting the inclusion of all six attributes. Finally, we emphasize that the main quantitative results reported in Table 1 are computed on the full test set (N = 139) without any threshold-based filtering, ensuring that the overall performance conclusions are not subject to selection bias.
>
> **Q4 (Weakness 4):** The paper would benefit from more qualitative visualization to assess boundary behavior and clinical plausibility.
>
> **A4:** We thank the reviewer for this valuable suggestion and agree that qualitative visualization is essential for evaluating boundary accuracy in CTV segmentation. To address this, we have expanded the qualitative results in **Appendix B.3.1** with revised figures from representative test cases. The updates include two key improvements.
>
> **1.Zoom-in boundary analysis.**
> We add synchronized ROI zoom-in views to enable detailed inspection of critical boundaries. Vision-only methods (U-Net, nnU-Net, UNETR) capture coarse CTV extents but often produce over-smoothed or inaccurate boundaries in anatomically complex regions. Representative text-conditioned methods (BiomedParse, SAT) show limited improvement, indicating that direct image-text fusion is insufficient for precise boundary control. In contrast, ReaCT better preserves fine anatomical details and yields more clinically plausible contours.
>
> **2.Diverse pathological cases.**
> The selected cases span diverse clinical conditions, including different Gleason scores and varied surgical margin locations. Across these scenarios, ReaCT consistently adapts boundary placement according to pathology while maintaining stable and realistic CTV shapes.
>
> Overall, the enhanced visualizations show that ReaCT improves boundary adherence and clinical plausibility beyond quantitative metrics.
>
> **Due to the character limit, please refer to the continuation below.**

---

> ### Author Response · Authors · 2026-01-25
> **Response 3/3**
>
> **Q5 (Comments 1):** A brief discussion on how the framework could adapt to evolving or institution-specific clinical guidelines would strengthen the clinical applicability.
>
> **A5:** We thank the reviewer for this valuable suggestion and agree that adaptability to evolving clinical guidelines is crucial for real-world deployment. ReaCT explicitly separates the Guideline-Informed LLM Agent from the segmentation network, ensuring that guideline-specific knowledge is not implicitly encoded in model weights but instead encapsulated within flexible LLM prompts. As clinical guidelines evolve or institutional practices differ, adaptation can be achieved by updating the attribute definitions or the guideline-to-attribute mappings, without modifying the core segmentation model. This modular design enables ReaCT to accommodate guideline updates and institutional variations in a transparent and controllable manner. As suggested, we have added a dedicated discussion of this aspect in **Section 3.4** of the revised manuscript.
>
>
> **Q6 (Comments 2):** Providing more implementation details or pseudocode for the LLM agent pipeline could improve reproducibility and clarity for readers.
>
> **A6:** We thank the reviewer for this helpful suggestion regarding reproducibility and clarity. We would like to clarify that the implementation details of the LLM-based pipeline were already described in the original submission (Appendix B). To further improve clarity and reproducibility, we have now added an explicit pseudocode description of the full LLM-based attribute extraction pipeline in the revised **Appendix B**. The added algorithm box provides a concise, end-to-end overview of the pipeline.
> In addition, following Reviewer tRf9’s suggestion, we refined the terminology for conceptual precision. While the LLM component performs guideline-informed reasoning by interpreting free-text pathology reports in a schema-constrained manner, it operates as a feed-forward module rather than an interactive agent under most contemporary definitions of agentic AI. We therefore adopt the term "Guideline-Informed Attribute Extractor" throughout the revised manuscript to avoid conceptual overstatement.
> We believe these additions substantially improve the transparency, clarity, and reproducibility of the proposed framework.

---

### Official Review · Reviewer_tRf9 · 2026-01-07

**Confidence:** 3
**Preliminary Rating:** 4
**Final Rating:** 4

**Summary:**

This paper addresses automatic clinical target volume (CTV) segmentation for post-operative prostate cancer radiotherapy, a clinically important yet challenging task because the CTV is not directly visible on CT and must be inferred from pathology findings and clinical guidelines. Existing segmentation methods largely rely on visual cues or coarse text conditioning, which are insufficient to capture the guideline-driven, pathology-dependent spatial reasoning used by clinicians. To tackle this, the authors propose ReaCT, a framework that reformulates CTV segmentation as a multimodal reasoning problem combining imaging, pathology reports, and guideline knowledge via large language models.

**Strengths:**

- The design is clear: Explicit decomposition into guideline-informed attribute extraction and attribute-specific reasoning.
- Strong empirical gains over both vision-only and coarse text-conditioning baselines.

**Weaknesses:**

- All experiments are conducted on a single-center in-house dataset. CTV delineation is known to vary substantially across institutions and guideline interpretations. It remains unclear whether the learned reasoning generalizes beyond this specific clinical practice.
- The “LLM Agent” in ReaCT functions as a static one-shot preprocessing pipeline rather than an interactive, adaptive, or decision-making agent. Actually, it seems that no interaction with the segmentation model: The LLM agent is fully decoupled from the segmentation network; No refinement if segmentation appears inconsistent with extracted attributes. Thus, calling it an agent is conceptually overstated under most contemporary definitions of agentic AI.

**Detailed Comments:**

-	The choice of six pathology attributes is guideline-driven but somewhat ad hoc; no analysis is given on whether fewer or alternative attributes suffice.
-	Computational cost and inference latency of the full pipeline are not discussed.

**Justification Of Final Rating:**

I think the paper addresses a relevant and underexplored clinical problem and proposes a principled multimodal reasoning framework that aligns well with real-world radiotherapy workflows. Overall, the rebuttal addresses my concerns, and especially be careful with the overstatement in the original manuscript. I maintain the acceptance score.

**Justification Of The Preliminary Rating:**

I think the paper addresses a relevant and underexplored clinical problem and proposes a principled multimodal reasoning framework that aligns well with real-world radiotherapy workflows. Yet, some concerns remain (see above), and some terms may be overstated.

**Questions To Address In The Rebuttal:**

See the above questions.

---

> ### Author Response · Authors · 2026-01-25
> **Response 1/2**
>
> We thank the reviewer for the constructive feedback. All concerns have been addressed, and revisions are highlighted in yellow in the revised manuscript.
>
> **Q1 (Weakness 1):**  All experiments are conducted on a single-center in-house dataset. CTV delineation is known to vary substantially across institutions and guideline interpretations. It remains unclear whether the learned reasoning generalizes beyond this specific clinical practice.
>
> **A1:** We acknowledge the concern regarding the use of a single-center in-house dataset and the known inter-institutional variability in postoperative CTV delineation. Importantly, ReaCT is explicitly designed to mitigate such variability by conditioning segmentation on explicit pathological attributes and guideline-driven rules, rather than implicitly learning institution-specific contouring styles from images alone. In contrast to vision-only methods that are prone to overfitting local annotation habits, the reasoning in ReaCT focuses on pathology-dependent spatial decisions (e.g., margin status or extraprostatic extension) that are defined by consensus guidelines and shared across institutions. As a result, the learned reasoning is not tied to a specific clinical practice but is intended to generalize across institutions once corresponding multimodal inputs are available. We have clarified this design rationale and its implications for generalization in **Section 3.4** of the revised manuscript.
>
> **Q2 (Weakness 2):** The "LLM Agent" in ReaCT functions as a static one-shot preprocessing pipeline rather than an interactive, adaptive, or decision-making agent. Actually, it seems that no interaction with the segmentation model: The LLM agent is fully decoupled from the segmentation network; No refinement if segmentation appears inconsistent with extracted attributes. Thus, calling it an agent is conceptually overstated under most contemporary definitions of agentic AI.
>
> **A2:** We thank the reviewer for the clarification. We acknowledge that the term "agent" may be overstated under contemporary definitions of agentic AI, which typically emphasize interactive decision-making and iterative refinement loops. In our framework, the LLM component functions as a guideline-informed reasoning module that performs structured attribute extraction from pathology reports based on clinical guidelines. While it does perform reasoning (interpreting free-text reports against guideline criteria), it operates in a feed-forward manner without closed-loop interaction with the segmentation network.
> We have revised the manuscript to use more precise terminology, replacing "LLM Agent" with "**Guideline-Informed Attribute Extractor**" throughout the text to avoid conceptual overstatement. We thank the reviewer for helping us improve the clarity of our presentation. We also appreciate that extending ReaCT to incorporate iterative refinement (e.g., LLM verifying segmentation consistency with extracted attributes) is an interesting direction for future work.
>
> **Due to the character limit, please refer to the continuation below.**

---

> ### Author Response · Authors · 2026-01-25
> **Response 2/2**
>
> **Q3 (Comment 1)**: The choice of six pathology attributes is guideline-driven but somewhat ad hoc; no analysis is given on whether fewer or alternative attributes suffice.
>
> **A3:** We thank the reviewer for this helpful comment. We clarify that the six pathological attributes constitute a clinically sufficient set to operationalize consensus postoperative CTV guidelines (e.g., ESTRO ACROP, RTOG), and that their selection is neither ad hoc nor redundant. We support this claim from both theoretical and empirical perspectives.
>
> **1.Theoretical necessity.** The selected attributes are derived directly from established clinical guidelines and are further confirmed by our radiation oncologist co-authors. Each attribute maps to a specific decision node in the standard postoperative CTV delineation workflow. For example, seminal vesicle invasion (SVI) determines the superior CTV extent, while extraprostatic extension (EPE) and surgical margin status govern localized expansion into high-risk periprostatic regions. Lymph node status further determines whether the target volume is restricted to the prostate bed or extended to include pelvic nodal regions. Removing any of these attributes would prevent faithful implementation of guideline-defined boundary rules.
>
> **2. Empirical validation.** Our results show that the selected attributes are both necessary and complementary. As shown in Fig. 2(a), most patients (87.8%) benefit from combining multiple attributes rather than any single one, indicating heterogeneous, attribute-dependent effects. Fig. 2(b) further shows that different attributes affect partially distinct patient subsets (e.g., Stage vs. Surgical Margin). As suggested, we additionally performed a leave-one-out ablation in **Appendix B.3.4** of the revised manuscript, removing each attribute in turn while keeping all other components fixed. Removing any single attribute consistently degrades performance, confirming that the attributes are jointly necessary rather than redundant.
>
> | Model Variant                    | Dice ↑              | HD95 (mm) ↓          | ASSD (mm) ↓          |
> |----------------------------------|---------------------|----------------------|----------------------|
> | ReaCT (6 attrs)                  | 0.8185 ± 0.05       | 4.15 ± 1.66          | 1.38 ± 0.48          |
> | w/o Stage                        | 0.8092 ± 0.05 (-0.93%) | 4.48 ± 1.71 (+0.33) | 1.47 ± 0.50 (+0.09) |
> | w/o Gleason Score                | 0.8101 ± 0.05 (-0.84%) | 4.44 ± 1.70 (+0.29) | 1.46 ± 0.49 (+0.08) |
> | w/o Extraprostatic Extension     | 0.8048 ± 0.05 (-1.37%) | 4.62 ± 1.76 (+0.47) | 1.51 ± 0.51 (+0.13) |
> | w/o Seminal Vesicle Invasion     | 0.8062 ± 0.05 (-1.23%) | 4.55 ± 1.74 (+0.40) | 1.49 ± 0.50 (+0.11) |
> | w/o Surgical Margin              | 0.8019 ± 0.05 (-1.66%) | 4.82 ± 1.89 (+0.67) | 1.56 ± 0.52 (+0.18) |
> | w/o Lymph Node Status            | 0.8085 ± 0.05 (-1.00%) | 4.41 ± 1.69 (+0.26) | 1.45 ± 0.49 (+0.07) |
>
> Taken together, these results confirm that the selected attributes are jointly necessary for robust postoperative CTV segmentation.
>
> **Q4 (Comment 2):** Computational cost and inference latency of the full pipeline are not discussed.
>
> **A4:** We thank the reviewer for highlighting the need to clarify the computational cost of the full ReaCT pipeline. These details have been explicitly addressed in **Appendix B.3.3** of the revised manuscript. The computational workflow of ReaCT is designed for efficiency through a two-stage process:
> 1. Offline attribute extraction. The guideline-informed LLM module is executed once during preprocessing to extract structured pathological attributes from clinical reports. In our implementation, this step takes approximately 2-5 seconds per case using GPT-4o (including model inference and API overhead), and the extracted attributes are cached for all subsequent segmentation runs. Importantly, this preprocessing step is fully decoupled from the segmentation network and does not contribute to the runtime or GPU memory footprint during segmentation inference.
> 2. Segmentation Network. The segmentation stage consists of a single forward pass through the multimodal network, with an inference latency of approximately 85 ms per case for a 320×320×64 volumetric input. Peak GPU memory usage during inference is approximately 27 GB in FP32. In practical clinical settings, inference is typically performed using half-precision (FP16), which reduces the memory footprint of the frozen 7B backbone by approximately 2×, lowering the requirement to around 13-14 GB and enabling deployment on standard workstation GPUs.  Finally, postoperative CTV delineation is an offline treatment planning task in clinical practice, typically requiring 20-30 minutes of clinician time. Therefore, the reported inference latency is negligible in comparison and supports seamless integration into existing PACS/TPS workflows.

---

### Official Review · Reviewer_zMdu · 2026-01-09

**Confidence:** 3
**Preliminary Rating:** 3
**Final Rating:** 4

**Summary:**

The authors propose ReaCT, a framework for postoperative prostate CTV segmentation that integrates patient-specific pathological attributes extracted from clinical reports. The pipeline uses a GPT-4o agent to distill these attributes, which are then processed by a "Multimodal Reasoner" (3D ViT + LLaMA-2 backbone) and fused into a standard 3D U-Net via attribute-specific query tokens. Evaluation is performed on a private dataset of 688 post-operative prostate cancer patients.

**Strengths:**

* The authors propose to explicitly incorporate pathological attributes for the CTV segmentation process, mirroring the clinical workflow of radiation oncologists. This transforms an often ill-defined segmentation task (segmenting the postoperative prostate bed) into a solvable, rule-based problem by integrating specific non-visual risk factors like the Gleason score.
* The work uses attribute-specific query tokens (SEG) to disentangle the spatial influence of distinct risk factors. This might avoid the information bottleneck of global text pooling and allow the model to apply specific spatial modulations corresponding to clinical rules (e.g. “confirmed seminal vesicle invasion mandates the inclusion of the entire seminal vesicle bed”).
* The method demonstrates improved robustness over vision-only baselines in scenarios with limited training data (5-20%), suggesting that semantic priors help stabilize learning when training data is sparse.

**Weaknesses:**

* The experimental results undermine the paper's core hypothesis. The w/o Textual Tokens ablation, which uses the large MLLM backbone only on visual features, achieves a Dice score of 0.8015. This outperforms the standard nnU-Net (0.7822) and even the text-prompted baselines. This suggests that the performance gains are driven by the huge parameter capacity of the hybrid architecture (U-Net + ViT + LLaMA), rather than the proposed "multimodal reasoning." If the model works best without text, the reasoning component is likely unneeded.
* The proposed architecture is computationally expensive and complex. To justify this, the authors must compare it against a simple, resource-efficient baseline, e.g. a standard 3D U-Net with clinical attributes concatenated as one-hot encoded vectors. The absence of this baseline makes it difficult to determine if the heavy LLaMA-2 backbone provides any real benefit over simple metadata fusion.
* It is unclear how the “w/o Textual Tokens” ablation is implemented If the attribute tokens are removed, what serves as the query input to the LLaMA backbone?
* The entire pipeline depends on a GPT4 agent to extract accurate attributes from reports, yet the accuracy of this extraction is never validated. There is no metric (e.g F1 score) reporting how often the agent hallucinates or misinterprets clinical data. Any error here propagates to the segmentation. this failure mode is unquantified.
* The paper lacks any reports on the training and inference latency (training time, inference time, VRAM for inference). Given the complex and compute heavy architecture, it is probably cannot be easily integrated into a clinical workflow.
* The paper frequently claims "significant" improvements without providing statistical tests (e.g., p-values). Additionally, the reliance on a private in-house dataset limits reproducibility and prevents benchmarking for the broader community.

**Detailed Comments:**

The terminology in the method section should also be used in Figure 1 to better follow the depicted process.

**Justification Of Final Rating:**

The authors have provided a thorough rebuttal. The additional experiments and the revised manuscript address most of my concerns, warranting no further questions from my side. Acknowledging the increased quality of the revised work, I will raise my score accordingly.

**Justification Of The Preliminary Rating:**

While the clinical problem is relevant and the "query token" design is interesting, the paper fails to convincingly validate its core contribution. The ablation studies paradoxically show that the text-free version of the model outperforms standard baselines, suggesting that the reported gains might stem from the massive increase in model parameters (ViT + LLaMA + U-Net) rather than the proposed "guideline-informed reasoning." Furthermore, the omission of a simple metadata-concatenation baseline makes the complexity of the proposed solution unjustified. Finally, the work lacks statistical rigor and leaves the attribute extraction via GPT4 agent unverified.

**Questions To Address In The Rebuttal:**

* Why does the w/o Textual Tokens variant (Dice 0.8015) considerably outperform the optimized nnU-Net (0.7822)? Does this not indicate that your gains come from model capacity rather than multimodal reasoning?
* Why is there no comparison against a standard U-Net with simple attribute concatenation (one-hot encoding)? How do you justify the cost of a 7B parameter LLM without this baseline?
* What is the accuracy/F1 score of the GPT-4o attribute extraction compared to human experts? How do you ensure that this agent is not the “point of failure” for the framework’s predictions?
* Please provide p-values for the comparisons in Table 1 to support claims of "state-of-the-art" performance and “significant” improvements.
* Please provide details on training time, inference time, and VRAM. Assess the potential challenges that might occur when adapting the model in a clinical workflow.

---

> ### Author Response · Authors · 2026-01-25
> **Response 1/3**
>
> We thank the reviewer for the time and effort spent carefully evaluating our work. We have addressed all concerns raised and incorporated corresponding revisions in the revised manuscript. All changes are highlighted in yellow for ease of reference.
>
> **Q1 (Weakness 1 / Question 1):**  The w/o Textual Tokens variant significantly outperforms nnU-Net. Does this indicate that the gains mainly come from increased model capacity rather than attribute-guided multimodal reasoning, and how does this support the core claim of ReaCT?
>
> **A1:** We thank the reviewer for this question. We agree that the strong performance of the w/o Textual Tokens variant requires careful interpretation and have clarified this point in **Section 3.3.1** of the revised manuscript. Importantly, this result does not contradict our hypothesis.
>
> First, the performance gain over nnU-Net is not due to parameter count alone, but to semantic priors transferred from large-scale language pretraining. As shown in recent work [1], frozen LLM layers can act as semantic-aware visual boosters, improving global visual representations even when operating on visual tokens only. This explains why the vision-only LLM variant outperforms CNN-based baselines such as nnU-Net.
> Second, textual attributes remain essential for attribute-conditioned spatial reasoning. When visual tokens are removed (w/o Visual Tokens), our framework still outperforms representative text-conditioned baselines (e.g., BiomedParse [2], SAT [3]), indicating that the autoregressive MLLM architecture more effectively models implicit pathological semantics than fixed text encoders. Moreover, as shown in Appendix Table 2, corrupting individual attributes leads to systematic, attribute-specific performance degradation, demonstrating explicit reliance on attribute-conditioned reasoning rather than generic model capacity.
> Overall, these results indicate that while semantic priors improve visual representations, multimodal, attribute-guided reasoning is essential when CTV delineation requires pathology-dependent spatial decisions that cannot be resolved from imaging alone.
>
> **Q2 (Weakness 2 / Question 2):**  Lack of comparison with a simple baseline (e.g., 3D U-Net + one-hot attributes). Benefits of the 7B LLM are unclear.
>
> **A2:**
> We thank the reviewer for this important suggestion. To justify the use of a large LLM backbone, we have added two simple and resource-efficient metadata fusion baselines to **Table 1**, with corresponding analysis in **Section 3.2**. Specifically, we include
> (1) a 3D U-Net with one-hot encoded clinical attributes, and
> (2) a 3D U-Net with concatenated text embeddings extracted by a pretrained biomedical text encoder (PubMedBERT).
> Both baselines follow the same fusion strategy as the CLIP-Driven Universal Model [4], concatenating attribute representations with global visual features obtained via global average pooling. The results are summarized below.
> | Method | Dice ↑ | HD95 (mm) ↓ | ASSD (mm) ↓ |
> |------|--------|-------------|-------------|
> | 3D U-Net (Vision Only) | 0.7847 ± 0.01 | 6.97 ± 2.33 | 2.13 ± 0.54 |
> | 3D U-Net + Text Embeddings (Concat) | 0.7675 ± 0.02 | 7.85 ± 2.10 | 2.45 ± 0.65 |
> | 3D U-Net + One-Hot Encoding | 0.7881 ± 0.02 | 6.84 ± 1.95 | 2.03 ± 0.45 |
> | **ReaCT (Ours)** | **0.8185 ± 0.05** | **4.15 ± 1.66** | **1.38 ± 0.48** |
>
> Based on the results, simple metadata fusion does not effectively improve postoperative CTV segmentation. Direct concatenation of text embeddings even degrades performance compared to the vision-only baseline, a trend also observed in representative text-conditioned methods such as SAT [3]. This suggests that naively fusing high-dimensional textual representations with visual features introduces a modality gap that limits the effective use of pathological information.
> One-hot encoding yields only a marginal improvement (+0.34% Dice). While computationally efficient, it treats clinical attributes as independent symbols and fails to capture the structured and correlated relationships defined by clinical guidelines (e.g., Stage T3b implying seminal vesicle invasion and increased extraprostatic extension risk).
>
> In contrast, ReaCT substantially outperforms both one-hot encoding (+3.04% Dice) and concatenated text embeddings (+5.1% Dice). By jointly processing image tokens and pathological attributes within a unified autoregressive sequence, ReaCT explicitly models attribute-conditioned spatial dependencies rather than performing static feature fusion. This design enables contextual, guideline-driven reasoning grounded in image evidence, explaining why the observed gains cannot be attributed to feature augmentation alone.
>
> **Due to the character limit,  please refer to the continuation below.**

---

> ### Author Response · Authors · 2026-01-25
> **Response 2/3**
>
> **Q3 (Weakness 3):** In the w/o Textual Tokens ablation, when all attribute tokens are removed, what serves as the input or "query" to the LLaMA backbone?
>
> **A3:**   In this ablation, all attribute tokens are removed and the LLaMA backbone receives only visual tokens. Following M3D-LaMed [4], 3D image features extracted by the M3D-ViT encoder are compressed by a spatial pooling projector and directly used as the input sequence to LLaMA. In this vision-only setting, the visual tokens themselves constitute the input context and are processed via causal self-attention, without any textual or attribute-based queries. We have clarified this in **Section 3.3.1** of the revised manuscript.
>
> **Q4 (Weakness 4 / Question 3)**: The pipeline relies on GPT-4o for attribute extraction, but its accuracy is not validated. What is the extraction accuracy/F1 compared to human experts, and how is the extractor prevented from becoming a single point of failure?
>
> **A4:** We thank the reviewer for raising this concern. We have added a quantitative validation of attribute extraction in **Appendix B.3.1**.
> (1) Quantitative validation.
> We randomly sampled 200 cases and asked two radiation oncologists to verify the GPT-4o–extracted attributes against the original pathology reports. Disagreements were resolved by consensus. The extractor achieves high accuracy and F1 across all six attributes, indicating reliable performance:
> | Attribute | Accuracy (%) | F1-score |
> |----------|--------------|----------|
> | Stage | 97.5 | 0.96 |
> | Gleason Score | 96.3 | 0.95 |
> | Extraprostatic Extension | 97.0 | 0.96 |
> | Seminal Vesicle Invasion | 98.0 | 0.95 |
> | Surgical Margin | 95.0 | 0.94 |
> | Lymph Node Status | 99.0 | 0.97 |
> | **Overall** | **97.1** | **0.96** |
>
> (2) Robustness by design.
> ReaCT is explicitly designed to avoid treating the extractor as a single point of failure. First, the Guideline-Informed LLM Agent performs schema-constrained and guideline-grounded extraction, mapping pathology reports into a fixed set of clinically defined attributes rather than producing free-form text, which substantially reduces hallucination risk. Second, the extracted attributes are incorporated as soft conditioning signals rather than hard rules; visual features from CT images remain fully preserved and continuously contribute to segmentation through the multimodal reasoner. As a result, potential noise or errors in individual attributes do not catastrophically override visual evidence. Third, we empirically demonstrate that the model does not blindly rely on textual attributes. As shown in Appendix Table 2, systematically corrupting individual pathological attributes leads to attribute-dependent and gradual performance degradation, rather than abrupt failure. This behavior indicates that the model leverages attributes in a controlled and interpretable manner instead of treating them as absolute constraints.
> Together, these results demonstrate both high extraction reliability and robustness to rare extraction errors.
>
> **Q5 (Weakness 5 / Question 5):**  The paper does not report training or inference efficiency (runtime, VRAM), raising concerns about the clinical feasibility of this computationally heavy architecture. What are the training time, inference latency, and memory requirements, and how practical is ReaCT for clinical deployment?
>
> **A5:** We thank the reviewer for raising this important practical concern regarding computational cost and clinical integration. We have conducted a systematic benchmarking of training and inference efficiency on a single NVIDIA  H100 GPU.  The detailed results  have been added to **Appendix B.4** of the revised manuscript. The key model statistics are summarized below:
> | Metric             | Value            |
> |--------------------|------------------|
> | Total Parameters   | 7.02 Billion     |
> | Training Time      | ~2.75 hours      |
> | Inference Latency  | ~85 ms / case    |
> | Inference VRAM     | ~27 GB (FP32)    |
>
> While the peak VRAM usage of ~27 GB in FP32 reflects our research setting, ReaCT remains feasible for clinical deployment. In practice, FP16 inference reduces memory usage by ~2× to 13–14 GB, fitting standard workstation GPUs (e.g., RTX 3090/4090). For more constrained hardware, established INT8/INT4 quantization can further reduce VRAM to <10 GB with minimal accuracy impact, as shown in prior LLM compression studies [5]. Moreover, postoperative CTV delineation is an offline planning task (20–30 min clinically) [6], making the ~85 ms inference latency negligible and fully compatible with existing PACS/TPS workflows.
>
> **Due to the character limit,  please refer to the continuation below.**

---

> ### Author Response · Authors · 2026-01-25
> **Response 3/3**
>
> **Q6 (Weakness 6 / Question 4):**
> The paper claims "significant" improvements without reporting statistical tests, and relies on a private dataset, raising concerns about statistical validity and reproducibility. Can the authors provide p-values for Table 1 and clarify the reproducibility implications?
>
> **A6:** We thank the reviewer for highlighting the need for clearer statistical validation and reproducibility discussion. We address these concerns as follows.
>
> **(1) Statistical significance.**
> We conducted a Wilcoxon signed-rank test between ReaCT and the strongest baseline (w/o Textual Tokens) on the test set (N = 138). Improvements in Dice, HD95, and ASSD are all statistically significant. As requested, the corresponding p-values have been added to the footnote of **Table 1** in the revised manuscript. A summary is provided below.
> | Metric | Improvement | P-value | Significance |
> |------|-------------|---------|--------------|
> | Dice | +1.70% | 0.0033 | p < 0.01 |
> | HD95 | −0.37 mm | < 0.001 | p < 0.001 |
> | ASSD | −0.13 mm | < 0.001 | p < 0.001 |
>
> **(2) Private dataset and reproducibility.** We acknowledge the limitation of using a single-center private dataset. Large-scale public datasets for postoperative prostate CTV segmentation are scarce, and existing benchmarks typically lack paired pathology reports, which are essential for this task. Postoperative CTV delineation inherently requires correlating imaging with patient-specific pathological attributes (e.g., Gleason score, margin status), making multimodal data a necessity rather than an option. To address this, we rely on a comprehensive in-house cohort of 688 cases with both volumetric imaging and structured pathology, which is comparable in scale to commonly used radiotherapy benchmarks. While raw data cannot be publicly released due to privacy constraints, we commit to releasing our code, model checkpoints, and preprocessing pipelines upon acceptance. This limitation and its implications are discussed in **Section 3.4** of the revised manuscript.
>
> **Q7 (Comment 1)**: The terminology in the method section should also be used in Figure 1 to better follow the depicted process
>
> **A7:** We thank the reviewer for pointing out the terminology inconsistency between the method section and Figure 1. Following the reviewer’s suggestion, we revised **Figure 1** to fully align with the updated terminology and definitions in the manuscript. Specifically, we have replaced ambiguous labels such as "text sequence" with "input sequence" to better reflect the actual multimodal input composition, and renamed SEG token embeddings” as “attribute-specific reasoning embeddings” to clearly indicate their functional role in the proposed reasoning framework.
>
>
> **References:**
>
> [1] Tang, Fenghe, et al. "Pre-trained llm is a semantic-aware and generalizable segmentation booster." International Conference on Medical Image Computing and Computer-Assisted Intervention. Cham: Springer Nature Switzerland, 2025.
>
> [2] Zhao, Theodore, et al. "A foundation model for joint segmentation, detection and recognition of biomedical objects across nine modalities." Nature methods 22.1 (2025): 166-176.
>
> [3] Zhao, Ziheng, et al. "Large-vocabulary segmentation for medical images with text prompts." NPJ Digital Medicine 8.1 (2025): 566.
>
> [4] Bai, Fan, et al. "M3d: Advancing 3d medical image analysis with multi-modal large language models." arXiv preprint arXiv:2404.00578 (2024).
>
> [5] Lin, Ji, et al. "Awq: Activation-aware weight quantization for on-device llm compression and acceleration." Proceedings of machine learning and systems 6 (2024): 87-100.
>
> [6] Cha, Elaine, et al. "Clinical implementation of deep learning contour autosegmentation for prostate radiotherapy." Radiotherapy and Oncology 159 (2021): 1-7.

---

### Author Rebuttal · Authors · 2026-01-25

**Rebuttal:**

We thank all reviewers for their careful reading of our manuscript and for their insightful and constructive comments. We have carefully addressed each concern and revised the manuscript accordingly. All modifications, additional analyses, and clarifications have been incorporated into the revised version, with changes highlighted for ease of reference. We sincerely appreciate the reviewers’ time and effort, which have helped us improve the clarity, rigor, and presentation of our work.

**Supporting Material:**

/attachment/471288d88ab95aff43b3582e8a96f6c4448ca108.pdf

---

### Meta-Review · Area_Chair_atAK · 2026-02-09

**Recommendation:** Accept (Oral)
**Confidence:** 5

**Metareview:**

The proposed work shows technical merit and has been validated through thorough experimentation. All reviewers have agreed to accept it, and I agree with the reviewers.

---

### Decision · Program_Chairs · 2026-02-13

Accept (Poster)